# A Partition Cover Approach for Tokenization

**Jia Peng Lim**
Singapore Management University
`jiapeng.lim.2021@phdcs.smu.edu.sg`

**Shawn Tan**
MIT-IBM Watson AI Lab
`shawntan@ibm.com`

**Davin Choo**[*][†]
Harvard University
`davinchoo@seas.harvard.edu`

**Hady W. Lauw**[*]
Singapore Management University
`hadywlauw@smu.edu.sg`

## Abstract

Tokenization is the process of encoding strings into tokens of a fixed vocabulary size, and is widely utilized in Natural Language Processing applications. The leading tokenization algorithm today is Byte-Pair Encoding (BPE), which formulates the tokenization problem as a compression problem and tackles it by performing sequences of merges. In this work, we formulate tokenization as an optimization objective, show that it is NP-hard via a simple reduction from vertex cover, and propose a polynomial-time greedy algorithm GREEDTOK. Our formulation naturally relaxes to the well-studied weighted maximum coverage problem which has a simple $(1 - 1/e)$-approximation algorithm GREEDWMC. Through empirical evaluations on real-world corpora, we show that GREEDTOK outperforms BPE and UNIGRAM on compression and achieves a covering score comparable to GREEDWMC. Finally, our extensive pre-training for two transformer-based language models with 1 billion parameters, comparing the choices of BPE and GREEDTOK as the tokenizer, shows that GREEDTOK achieves a lower bit per byte even when we control for either the total dataset proportion or total training tokens.

## 1 Introduction

Tokenization encodes text into tokens from a fixed vocabulary and is fundamental to Natural Language Processing (NLP) applications. With Large Language Models (LLMs) growing in prominence, understanding tokenization has become increasingly important, as it plays an integral role in their architectures and even modest gains in efficiency can yield substantial computational savings. Furthermore, LLMs such as LLaMA [TLI+23] and Mistral [JSM+23] use fixed-length context windows, which will benefit from tokenizers with better compression utility that enables them to fit more information within their context window. There are also prompt-based [WWS+22, YYZ+23] and fine-tuning [FTL+23] techniques that increase the number of processed tokens to improve model performance.

A common way to formalize the tokenization problem is as a compression task that minimizes the ratio of tokens produced when tokenizing the input data. The leading tokenization algorithm today is Byte-Pair Encoding (BPE) [Gag94, SHB16, KHM+23], which formulates the tokenization problem as a compression problem and tackles it by performing a sequence of pairwise merges. Due to its popularity, there have been a multitude of recent works analyzing the theoretical properties of BPE [ZMG+23, KV24, WBP24]. Another approach frames tokenization as a pathing/sequence problem

---

[*]Equal Advising.

[†]Part of the work was done while the author was affiliated with the National University of Singapore, Singapore

39th Conference on Neural Information Processing Systems (NeurIPS 2025).

[Kud18, SRZ$^+$24], Unigram [Kud18] is one such example and is favored by models with bidirectional contexts, i.e. [YDY$^+$19] and [RSR$^+$20]. Although UNIGRAM is believed to perform better in pre-training tasks [BD20], its compression performance is weaker compared to BPE [SRZ$^+$24], affecting its adoption for large-scale expensive pre-training in language modeling.

**Contributions.** In this work, we deviate from the usual path, sequence, and merge-based tokenization formulations. Instead, we propose to examine the tokenization problem as a problem of *covering*.

**C1. Partition cover formulation.** We introduce a partition cover optimization formulation of the tokenization problem that goes beyond prior merge-based approaches [ZMG$^+$23, KV24]. These earlier methods rely on bottom-up pairwise merges from an existing token set. However, the idea of tokenization is simply to efficiently represent a corpus with a judiciously selected set of tokens, whose construction is *independent* of such merge patterns. Our formulation subsumes these prior formulations in the following sense: all merge-based solutions are valid solutions to our formulation, but a solution to our formulation need not be based on bottom-up pairwise merges.

**C2. Simple NP-hardness proof.** We provide a simple and intuitive proof that tokenization is NP-hard in our optimization objective formulation via a reduction from vertex cover [Kar72]. Although there has been a recent concurrent work [WBP24] that also showed that the tokenization problem is NP-hard, our proof is arguably simpler due to our formulation.

**C3. New polynomial-time tokenizer.** We propose a polynomial-time greedy algorithm GREEDTOK that does not rely on a sequence of pairwise token merges or path construction. Evaluation on four real-world corpora shows that GREEDTOK outperforms BPE and UNIGRAM with a stronger compression of ∼3% tokens per word. Our implementation and compression evaluations can be found at https://github.com/PreferredAI/pcatt/; see supplementary materials.

**C4. Downstream comparison.** Preliminary analysis revealed that GREEDTOK's token sets simultaneously attain BPE's compressibility and UNIGRAM's token quality. To empirically validate downstream performance, we pre-trained two transformer-based LLMs with 1 billion parameters that differ only in the use of GREEDTOK or BPE as the tokenizer algorithm. Our results show that GREEDTOK outperforms BPE in both common benchmark tasks and in bits per byte, even after controlling for either the total dataset proportion or total training tokens.

**C5. Empirical approximation of objective.** Our partition cover formulation naturally relaxes to the well-studied weighted maximum coverage problem [Kar72, CK08] which has a simple $(1 - 1/e)$-approximation algorithm GREEDWMC [Hoc96]. We empirically show that GREEDTOK and GREEDWMC achieve a comparable objective function value for large $k$, despite the latter being a relaxed problem. Although a formal approximation guarantee for GREEDTOK currently escapes us, our analysis holds for practical scenarios encountered by NLP practitioners and this empirical method of investigating approximation guarantees may be of independent interest.

**Related work.** There has been a recent surge of interest in analyzing tokenization. [ZMG$^+$23] initiated a formal study of BPE using a bottom-up tokenization problem formulation that restricts token construction to sequential merges of two tokens from the existing vocabulary. [KV24] proved that this bottom-up tokenization problem, and its more general variant,[3] is APX-complete using linear reduction [PY88] from maximum cut [Kar72] in cubic graphs. They also showed that BPE approximates a worst-case factor of between $0.333$ and $0.625$ for their general variant. In a recent concurrent work, [WBP24] showed that both the bottom-up tokenization problem formulation and our partition cover formulation are NP-complete from the reduction of the maximum 2-satisfiability problem. Beyond theory, empirical studies such as [LBM23] and [SFWN23] have examined the practical downstream impact of tokenizer selection in NLP tasks. With regard to tokenizer-free architectures [TTR$^+$22, YSF$^+$23, PPR$^+$24], our formulation draws the link to the fundamental binary decision problem of whether to merge adjacent characters, as one can view next-byte predictions as merging decisions.

Compared to prior and concurrent works, our NP-hardness proof (Theorem 1) is arguably simpler due to our formulation, and we contribute a novel tokenizer that is competitive in real-world scenarios.

**Outline of paper.** We give our partition cover optimization formulation in Section 2, we prove that it is NP-hard in Section 3. GREEDTOK is designed in Section 4 and Section 5 contains our empirical evaluation against real world corpora. Finally, we conclude with some discussions in Section 6.

---

[3]Referred to as optimal merge sequence and optimal pair encoding respectively in their work.

**Notation.** We use standard set notations such as $|\mathbf{A}|$ to represent the size of set $\mathbf{A}$, and standard asymptotic notations such as $O(\cdot)$. Numbers are represented with small letters, strings/words with capital letters, and sets with bold letters. Unordered sets are denoted by $\{\cdot\}$ and ordered tuples are denoted by $(\cdot)$. We describe words in plaintext, e.g., hello, or as a tuple of singletons, e.g., (h,e,l,l,o).

## 2  A general optimization formulation for the tokenization problem

Let $\mathbf{\Sigma}$ be a fixed alphabet , i.e., a basic character set. This may be the 26 lowercase English letters or the full Unicode set, depending on the context. A corpus $\mathcal{C} = (\mathbf{W}, \textsc{count})$ consists of a set of distinct words $\mathbf{W} \subseteq \mathbf{\Sigma}^+$, and a count function $\textsc{count} : \mathbf{W} \to \mathbb{N}$ that indicates word frequencies. Given a word $W \in \mathbf{\Sigma}^+$ and a set of tokens $\mathbf{S} \subseteq \mathbf{\Sigma}^+$, let $\textsc{partition}(W, \mathbf{S})$ denote the minimum number of tokens from $\mathbf{S}$ that can be concatenated in sequence to form $W$. For example, the word $W = \text{abc}$ can be tokenized as ab␣c but not as ac␣b, despite both token sets covering the same characters.

The goal of minimizing the total number of tokens used to represent a corpus is often referred to as optimizing compression utility.

**Problem 1** (Tokenization search problem $\textsc{Tok}$). Let $\mathbf{\Sigma}$ be a fixed alphabet and define the base token set $\mathbf{B} = \{(w) : w \in \mathbf{\Sigma}\}$ as all singleton characters. Given a corpus $\mathcal{C} = (\mathbf{W}, \textsc{count})$, a token budget $k \in \mathbb{N}_{>0}$, and a set of candidate tokens $\mathbf{T} \subseteq \mathbf{\Sigma}^+$, the goal is to find a subset $\mathbf{S} \subseteq \mathbf{T}$ such that $|\mathbf{S}| \leq k$ and $\sum_{W \in \mathbf{W}} \textsc{partition}(W, \mathbf{S} \cup \mathbf{B}) \cdot \textsc{count}(W)$ is minimized. Note that tokens $\mathbf{T}$ are candidate substrings drawn from $\mathbf{\Sigma}^*$ that may or may not correspond to actual words.

As in standard practice, we consider the corresponding decision variant to establish NP-hardness. Once the decision variant is shown to be NP-hard, the search problem inherits this hardness, since solving the search version yields a solution to the decision version by evaluating different thresholds (in our case, the value $\ell$).

**Problem 2** (Tokenization decision problem). With the same setup as Problem 1, and given an additional integer threshold $\ell \in \mathbb{N}_{>0}$, determine whether there exists a subset $\mathbf{S} \subseteq \mathbf{T}$ such that $|\mathbf{S}| \leq k$ and $\sum_{W \in \mathbf{W}} \textsc{partition}(W, \mathbf{S} \cup \mathbf{B}) \cdot \textsc{count}(W) \leq \ell$.

Our formulation differs subtly but significantly from prior tokenization models, such as those in [ZMG+23, KV24]. Rather than viewing tokenization through the lens of string compression algorithms, we reduce tokenization to the following question: *Are two adjacent singleton symbols within a string covered by the same token?* This perspective emphasizes that tokenization is fundamentally about selecting a compact vocabulary that fully covers the corpus without imposing algorithmic constraints such as bottom-up merge sequences, which are artifacts of specific approaches like BPE. We further elaborate on the implications of our formulation in Section 4.1.

## 3  Tokenization is NP-hard

In this section, we prove that the tokenization decision problem (Problem 2) is NP-hard by a reduction from the vertex cover problem, which is known to be NP-hard [Kar72]. Given a graph $\mathcal{G} = (\mathbf{V}, \mathbf{E})$ with vertex set $\mathbf{V}$ and edge set $\mathbf{E}$, a vertex cover is a subset $\mathbf{S} \subseteq \mathbf{V}$ of vertices such that $|\mathbf{S} \cap \{U, V\}| \geq 1$ for every edge $\{U, V\} \in \mathbf{E}$. The decision variant of the vertex cover problem asks whether there exists a subset $\mathbf{S}$ of size at most $k$.

**Theorem 1.** *The tokenization problem is NP-hard.*

*Proof.* We will prove this by reducing the vertex cover problem, which is known to be NP-hard [Kar72], to the tokenization problem. Given an arbitrary vertex cover problem instance, we show how to construct a corresponding tokenization instance. Then, we argue that the derived tokenization problem instance is a YES instance if and only if the original vertex cover problem instance is a YES instance. In this proof, for clarity, we will write words $W \in \mathbf{W}$ as a tuple of singletons instead of usual plaintext, e.g. (h,e,l,l,o) instead of hello.

**Construction.** Consider an arbitrary vertex cover problem given by the graph $\mathcal{G} = (\mathbf{V}, \mathbf{E})$ over $n$ vertices $\mathbf{V} = \{V_1, \ldots, V_n\}$ and a positive integer $k \in \mathbb{N}_{\geq 0}$. To construct an instance of the tokenization problem, we first define the alphabet as follows: $\mathbf{\Sigma} = \{V_1, \ldots, V_n, @\}$ where $@$ is

an additional symbol which we will use later. So, we have $\mathbf{B} = \{(V_1), \ldots, (V_n), (@)\}$. For each edge $\{V_i, V_j\} \in \mathbf{E}$ with $i < j$, we create a word $W_{i,j} = (@, V_i, @, V_j, @)$ and define the set of words as $\mathbf{W} = \{W_{i,j} : \{V_i, V_j\} \in \mathbf{E}\}$ where each word has count 1, i.e. $\text{COUNT}(W) = 1$ for all $W \in \mathbf{W}$. Then, we define the set of candidate tokens $\mathbf{T} = \{(@, V_i, @) : V_i \in \mathbf{V}\}$. Finally, we set $\ell = 3|\mathbf{W}| = 3|\mathbf{E}|$ and associate the parameter $k$ in the vertex cover problem instance to the corresponding parameter $k$ in the tokenization problem instance. One can check that this derived tokenization instance can be constructed in polynomial time.

**Observation.** Observe that every word $W \in \mathbf{W}$ has length 5 and each token in $\mathbf{S}$ has length 3, so $\text{PARTITION}(W, \mathbf{S} \cup \mathbf{B})$ will either be 3, when there is some token in $\mathbf{S}$ that is a contiguous subword of $W$, or 5 otherwise. For instance, given the word $W_{i,j} = (@, V_i, @, V_j, @)$, we have $\text{PARTITION}(W_{i,j}, \mathbf{S} \cup \mathbf{B}) = 3$ if and only if at least one of $(@, V_i, @)$ or $(@, V_j, @)$ is chosen in $\mathbf{S}$ (both could be in $\mathbf{S}$). Furthermore, since all words have count 1, the tokenization problem becomes finding $\mathbf{S} \subseteq \mathbf{T}$ such that $|\mathbf{S}| \leq k$ and $\sum_{W \in \mathbf{W}} \text{PARTITION}(W, \mathbf{S} \cup \mathbf{B}) \leq \ell = 3|\mathbf{W}|$.

**YES instance of tokenization problem to YES instance of vertex cover.** Suppose there exists a subset $\mathbf{S} \subseteq \mathbf{T}$ of tokens such that $|\mathbf{S}| \leq k$ and $\sum_{W \in \mathbf{W}} \text{PARTITION}(W, \mathbf{S} \cup \mathbf{B}) \leq \ell = 3|\mathbf{W}|$. Then, from the observation above, we know that this can only happen when $\text{PARTITION}(W, \mathbf{S} \cup \mathbf{B}) = 3$ for every $W \in \mathbf{W}$. This implies that for each word $W_{i,j}$, at least one of $(@, V_i, @)$ or $(@, V_j, @)$ is chosen in $\mathbf{S}$. Therefore, $\mathbf{S}_{\mathcal{G}} = \{V_i \in \mathbf{V} : (@, V_i, @) \in \mathbf{S}\}$ is a subset of size $|\mathbf{S}_{\mathcal{G}}| = |\mathbf{S}| \leq k$ and corresponds to a vertex cover of the original graph $\mathcal{G}$.

**YES instance of vertex cover to YES instance of tokenization problem.** Suppose the original vertex cover instance for graph $\mathcal{G} = (\mathbf{V}, \mathbf{E})$ has a vertex cover $\mathbf{S}_{\mathcal{G}}$ of size $|\mathbf{S}_{\mathcal{G}}| \leq k$. Then, let us define $\mathbf{S} = \{(@, V_i, @) \in \mathbf{\Sigma}^+ : V_i \in \mathbf{S}_{\mathcal{G}}\}$ as the set of chosen tokens of size $|\mathbf{S}| = |\mathbf{S}_{\mathcal{G}}| \leq k$. Since $\mathbf{S}_{\mathcal{G}}$ is a set cover for $\mathcal{G}$, by construction of $\mathbf{W}$, we see that $\text{PARTITION}(W, \mathbf{S} \cup \mathbf{B}) = 3$ for all $W \in \mathbf{W}$. Therefore, $\sum_{W \in \mathbf{W}} \text{PARTITION}(W, \mathbf{S} \cup \mathbf{B}) = 3|\mathbf{W}|$. $\qquad\square$

**Example 1.** Consider a vertex cover instance on a graph $\mathcal{G} = (\mathbf{V}, \mathbf{E})$ with vertices $\mathbf{V} = \{V_1, \ldots, V_5\}$ and edges $\mathbf{E} = \{\{V_1, V_2\}, \{V_1, V_4\}, \{V_1, V_5\}, \{V_2, V_3\}, \{V_2, V_4\}, \{V_3, V_5\}\}$ where the subset $\mathbf{S} = \{V_1, V_3, V_4\}$ is a vertex cover of size $|\mathbf{S}| = k = 3$. Fig. 1 illustrates the corresponding tokenization problem instance created according to the construction in the proof of Theorem 1.

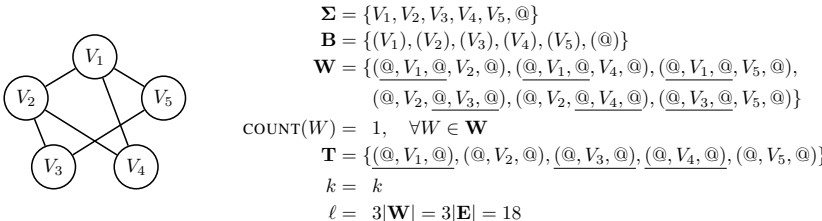

$$\mathbf{\Sigma} = \{V_1, V_2, V_3, V_4, V_5, @\}$$
$$\mathbf{B} = \{(V_1), (V_2), (V_3), (V_4), (V_5), (@)\}$$
$$\mathbf{W} = \{(@, \underline{V_1, @, V_2}, @), (@, \underline{V_1, @, V_4}, @), (@, \underline{V_1, @, V_5}, @),$$
$$(@, V_2, \underline{@, V_3, @}), (@, V_2, \underline{@, V_4, @}), (@, \underline{V_3, @, V_5}, @)\}$$
$$\text{COUNT}(W) = 1, \quad \forall W \in \mathbf{W}$$
$$\mathbf{T} = \{(\underline{@, V_1, @}), (@, V_2, @), (\underline{@, V_3, @}), (\underline{@, V_4, @}), (@, V_5, @)\}$$
$$k = k$$
$$\ell = 3|\mathbf{W}| = 3|\mathbf{E}| = 18$$

Figure 1: An example tokenization problem instance construction according to the proof of Theorem 1. The tokens corresponding to the vertex cover $\mathbf{S} = \{V_1, V_3, V_4\}$ are underlined in $\mathbf{T}$. A possible tokenization of $\mathbf{W}$ using $\mathbf{S} \cup \mathbf{B}$ is also given with tokens in $\mathbf{S}$ being underlined, showing that each word in $\mathbf{W}$ only needs 3 tokens.

## 4 GREEDTOK: Our greedy tokenizer

**Challenges in designing an efficient algorithm.** In Section 3, we showed that the tokenization problem (TOK) is NP-hard. Developing efficient algorithms for NP-hard problems typically involves strategies that trade off between exactness, runtime, and solution quality. Since our focus is on scalable, real-world applications, we aim for polynomial-time approximations and do not pursue fixed-parameter tractable algorithms. Unfortunately, the common approximation strategies that are used to design efficient algorithms for NP-hard problems with provable approximation guarantee are not applicable here. Firstly, while submodular functions admit efficient greedy $(1 - 1/e)$-approximations [NWF78], our objective is neither submodular nor supermodular (see Appendix A). Secondly, relax-and-round methods, like those used for vertex cover [WS11], become impractical due to the sheer scale of real-world corpora which induces large numbers of variables and constraints.

### 4.1 An equivalent mixed integer program formulation

To design our algorithm GREEDTOK for TOK, we begin by reformulating the problem in terms of a mixed linear program (MIP). This serves two purposes. First, the MIP provides a straightforward and intuitive framework that simplifies the definition and implementation of our greedy algorithm. Second, it naturally relaxes to the well-known weighted maximum coverage problem (WMC), which is submodular and admits a greedy $(1 - 1/e)$-approximation algorithm [Hoc96, Section 3.9]. Although we cannot formally establish approximation guarantees for GREEDTOK, its connection to WMC enables empirical comparisons with GREEDWMC in TOK instances; see Section 5.3.

We define COVER$(W, \mathbf{S})$ as the maximum number of adjacent singletons in word $W$ that can be grouped into tokens from $\mathbf{S}$, with each character used at most once. For example, with $W =$ scaredy and $\mathbf{S} = \{\text{care}, \text{edy}\}$, we have COVER$(W, \mathbf{S}) = 3$ from concatenating 3 adjacent singleton pairs in "care", constrained on the position of 'e' which can only be used once. Meanwhile, PARTITION$(W, \mathbf{S} \cup \mathbf{B}) = 4$ via s␣care␣d␣y. Notice that $|W| =$ PARTITION + COVER. This lets us rewrite the minimization objective from Problem 1 as an equivalent maximization objective:

$$\min \sum_{W \in \mathbf{W}} \text{COUNT}(W) \cdot \text{PARTITION}(W, \mathbf{S} \cup \mathbf{B}) = \max \sum_{W \in \mathbf{W}} \text{COUNT}(W) \cdot \text{COVER}(W, \mathbf{S}).$$

We refer to both forms as TOK. Now, recall that $\mathbf{W}$ represents the set of words in the corpus where each word $W = (W_1, \ldots, W_{|W|}) \in \mathbf{W}$ has length $|W|$ and appears with frequency COUNT$(W) \geq 1$. Although our formulation permits any candidate token set $\mathbf{T}$, identifying an optimal solution requires considering all substrings of length $\geq 2$ within $\mathbf{W}$, i.e. there is a total number of $|\mathbf{T}| \leq \sum_{W \in \mathbf{W}} \left( \binom{|W|}{2} - |W| \right)$ such substrings, where $\binom{|W|}{2}$ counts all start-end pairs, and we subtract $|W|$ to exclude singletons. In the following, we use the notation $A \subseteq B$ to denote that $A$ is a substring of $B$, e.g. for $\subseteq$ force, and adopt a 1-based indexing in the MIP below.

To formulate Problem 1 as an MIP, our goal is to choose a subset $\mathbf{S} \subseteq \mathbf{T}$ of size $|\mathbf{S}| \leq k$ such that the following objective is maximized, encoding $\max \sum_{W \in \mathbf{W}} \text{COUNT}(W) \cdot \text{COVER}(W, \mathbf{S})$, where $c_W = \text{COUNT}(W)$:

$$\max \sum_{W \in \mathbf{W}} c_W \cdot \left( \sum_{i=1}^{|W|-1} m_{i,i+1}^W \right) \tag{1}$$

with the binary variables $x_T \in \{0, 1\}$ for all tokens $T \in \mathbf{T}$ (*Did we choose token $T \in \mathbf{T}$, i.e. $T \in \mathbf{S}$?*), $m_{1,2}^W, \ldots, m_{|W|-1,|W|}^W \in \{0, 1\}$, for all words $W \in \mathbf{W}$ (*Are the $i^{th}$ singleton $W_i$ and the $(i+1)^{th}$ singleton $W_{i+1}$ covered by the same token?*), and $m_{1,2}^{W,T}, \ldots, m_{|W|-1,|W|}^{W,T} \in \{0, 1\}$, for all words $W \in \mathbf{W}$ and tokens $T \in \mathbf{T}$ (*Did token $T \in \mathbf{S}$ cover the $i^{th}$ singleton $W_i$ and the $(i+1)^{th}$ singleton $W_{i+1}$?*), under the following constraints:

- $\sum_T^{\mathbf{T}} x_T \leq k$.
- $x_T \geq m_{i,i+1}^{W,T}$ if $(W_i, W_{i+1}) \subseteq T$,
- $\sum_T^{\mathbf{T}} m_{i,i+1}^{W,T} \geq m_{i,i+1}^W$,
- $\sum_T^{\mathbf{T}} m_{i,i+1}^{W,T} \leq 1$,                                 $\forall W \in \mathbf{W}, \forall T \in \mathbf{T}, \forall i \in \{1, \ldots, |W|-1\}$.
- $m_{i,i+1}^{W,T} = m_{i+1,i+2}^{W,T}$ if $(W_i, W_{i+1}, W_{i+2}) \subseteq T$,    $\forall W \in \mathbf{W}, \forall T \in \mathbf{T}, \forall i \in \{1, \ldots, |W|-2\}$.
- $\sum_T^{\mathbf{T}} m_{s-1,s}^{W,T} \leq 1 - m_{s,s+1}^{W,T}$, if $(W_s, W_{s+1})$ starts $T$,  $\forall W \in \mathbf{W}, \forall T \in \mathbf{T}, \forall s \in \{2, \ldots, |W|-1\}$.
- $\sum_T^{\mathbf{T}} m_{e,e+1}^{W,T} \leq 1 - m_{e-1,e}^{W,T}$, if $(W_{e-1}, W_e)$ ends $T$,  $\forall W \in \mathbf{W}, \forall T \in \mathbf{T}, \forall e \in \{2, \ldots, |W|-1\}$.

For a more thorough explanation of our MIP formulation with examples, please refer to Appendix B.

### 4.2 Relation to weighted maximum coverage

Like the vertex cover problem, the weighted maximum coverage problem (WMC) is NP-hard [Kar72, WS11, Hoc96]. Given a set of elements $\mathbf{L} = \{L_1, \ldots, L_{|\mathbf{L}|}\}$ with weights $\mathcal{W} = \{w_1, \ldots, w_{|\mathbf{L}|}\}$, a collection of subsets $\mathbf{U} = \{U_1, \ldots, U_{|\mathbf{U}|}\}$ where each $U_i \subseteq \mathbf{L}$, and an integer budget $k$, the goal is to select $\mathbf{U}' \subseteq \mathbf{U}$, such that $|\mathbf{U}'| \leq k$, to maximize the total weights of covered elements $\sum_{L_i \in \bigcup \mathbf{U}'} w_i$.

With some effort, one can show that WMC admits a mixed integer program with the same objective as Eq. (1) but with fewer constraints. Details are provided in Appendix C.

**Implication.** Since WMC shares the same objective as TOK but under weaker constraints, its optimal value is at least that of TOK. As WMC admits a $(1 - 1/e)$-approximate greedy algorithm (GREEDWMC), this guarantee also applies to its performance in TOK instances, although the solution from GREEDWMC may violate tokenization constraints. Nevertheless, if GREEDTOK achieves objective values comparable to GREEDWMC, it suggests that GREEDTOK may offer a similar approximation ratio for TOK despite lacking formal guarantees.

### 4.3 A polynomial-time greedy algorithm

We now informally describe our algorithm, GREEDTOK, which consists of two main steps: (1) selecting a token set $\mathbf{S}$ from candidate substrings $\mathbf{T}$, and (2) tokenizing words $\mathbf{W}$ using $\mathbf{S}$; see Appendix G for pseudocode and examples.

We begin by constructing the candidate token set $\mathbf{T}$, considering all substrings of length $\geq 2$ within the words $\mathbf{W}$ in the corpus. Then, for any $\mathbf{S} \subseteq \mathbf{T}$, let $f(\mathbf{S})$ be the objective value in our MIP formulation (see Section 4.1). Starting with $\mathbf{S} = \emptyset$, we iteratively add tokens to $\mathbf{S}$ by selecting $\tau = \operatorname{argmax}_{T \in \mathbf{T} \setminus \mathbf{S}} f(\mathbf{S} \cup \{T\}) - f(\mathbf{S})$, subject to MIP constraints, to $\mathbf{S}$ until $|\mathbf{S}| = k$. This process induces a natural ordering within the tokens in $\mathbf{S}$.

To tokenize a word $W \in \mathbf{W}$ using $\mathbf{S}$, we scan its singletons to identify possible matches to the tokens in $\mathbf{S}$ and sort these matches by the order in which the tokens were added to $\mathbf{S}$. We then iterate through these candidate covers and, if the cover satisfies the MIP constraints, mark the corresponding positions in a bitmask $m^W$ to cover the substring with the selected token.

A direct implementation yields a runtime of $O(|\mathbf{T}| \cdot k \cdot \sum_{W \in \mathbf{W}} |W|)$ when selecting $\mathbf{S}$ and $O(|W|^2 \cdot \log |W|)$ when tokenizing a word $W$. Note that this token ordering arises from the greedy nature of GREEDTOK but is not required for solving TOK, just as merge sequences are not fundamental to tokenization. Despite the higher asymptotic costs than BPE, we show in Section 5.1 that with implementation optimizations, GREEDTOK is practical for real-world NLP use.

**Comparing to BPE.** GREEDTOK's token order resembles the merge sequence in BPE, as both select one token per iteration. However, GREEDTOK operates without the constraints of pairwise merges, allowing more flexible token selection. BPE beats GREEDTOK in terms of computational complexity, with a selection cost of $O(k \cdot \sum_{W \in \mathbf{W}} |W|)$ and per-word tokenization cost of $O(|W|^2)$ when using the pairwise caching approach [Ope23]. However, this selection cost is a one-off cost that does not affect downstream applications. Additionally, we empirically show that the modest overhead of $O(\log |W|)$ in tokenization is worth the improvements in downstream tasks; see Section 5.

**Comparing to UNIGRAM.** UNIGRAM's likelihood objective $\mathcal{L}$ can be interpreted as a negative log-weighted version of TOK; see derivation in Appendix D and example where optimizing $\mathcal{L}$ may yield unfavorable behavior. While both GREEDTOK and UNIGRAM freely select tokens from $\mathbf{T}$, UNIGRAM prunes $\mathbf{T}$ to size $|\mathbf{S}| = k$, while GREEDTOK builds $\mathbf{S}$ up from $\emptyset$. Computational complexity wise, UNIGRAM's selection cost of $O(|\mathbf{T}| \cdot \log k \cdot \sum_{W \in \mathbf{W}} |W|)$ beats GREEDTOK's but its per-word tokenization cost of $O(k \cdot |W|)$ exceeds GREEDTOK's when $k \gg |W| \log |W|$. With large $k$ being common in practical real-world use cases, this is one reason why UNIGRAM is often not used in production systems despite being known to produce higher-quality tokens [BD20, SRZ$^+$24].

## 5 Empirical evaluation of GREEDTOK on real-world datasets

Our implementation of GREEDTOK is on `C++` and accessible using `Python` bindings or through `HuggingFace`'s API via a simple import line, enabling easy integration onto existing codebases.

### 5.1 Evaluating GREEDTOK's compression

We compared GREEDTOK against BPE and UNIGRAM by measuring compression performance across four real-world corpora at varying token budget levels $|\mathbf{S}| = k$; see Table 1. We define the singleton set $\mathbf{B}$ as all 256 byte values, $\mathbf{W}$ as the set of space-delimited UTF-8 strings (byte sequences) extracted from each corpus, and introduce a special token to mark the start of a string following a

Table 2: This table reports the compression performance of GREEDTOK/BPE/UNIGRAM algorithms. For UNIGRAM, we increase the input $k$ by 75/84/108/94 respectively to account for the compulsory character inclusion into $\mathbf{T}$. GREEDTOK (GTK) outperforms BPE and UNIGRAM in larger corpora (arXiv, PubMed, wiki), with mean improvement of 2.88% over BPE and 3.43% over UNIGRAM.

| | $k$ | 1000 | 2000 | 3000 | 4000 | 5000 | | 2000 | 4000 | 6000 | 8000 | 10000 |
|---|---|---|---|---|---|---|---|---|---|---|---|---|
| GTK Tokens/Word | | 1.607 | 1.374 | 1.268 | 1.205 | 1.163 | | 1.603 | 1.397 | 1.301 | 1.244 | 1.206 |
| BPE Tokens/Word | UN | 1.688 | 1.431 | 1.311 | 1.241 | 1.194 | PubMed | 1.650 | 1.431 | 1.328 | 1.266 | 1.225 |
| GTK's Improvement (%) | | **4.86** | **3.99** | **3.33** | **2.92** | **2.54** | | **2.85** | **2.38** | **2.02** | **1.75** | **1.52** |
| UNIGRAM Tokens/Word | | 1.655 | 1.385 | 1.261 | 1.193 | 1.148 | | 1.699 | 1.465 | 1.359 | 1.297 | 1.257 |
| GTK's Improvement (%) | | **2.90** | **0.78** | -0.51 | -0.97 | -1.30 | | **5.63** | **4.63** | **4.21** | **4.05** | **4.02** |
| GTK Tokens/Word | | 1.742 | 1.475 | 1.349 | 1.275 | 1.226 | | 1.692 | 1.489 | 1.389 | 1.326 | 1.283 |
| BPE Tokens/Word | arχiv | 1.837 | 1.551 | 1.407 | 1.320 | 1.263 | wiki | 1.731 | 1.519 | 1.413 | 1.347 | 1.301 |
| GTK's Improvement (%) | | **5.12** | **4.94** | **4.15** | **3.41** | **2.89** | | **2.26** | **1.98** | **1.71** | **1.53** | **1.37** |
| UNIGRAM Tokens/Word | | 1.793 | 1.558 | 1.444 | 1.378 | 1.332 | | 1.793 | 1.558 | 1.444 | 1.378 | 1.332 |
| GTK's Improvement (%) | | **6.74** | **4.92** | **4.30** | **3.96** | **3.88** | | **5.62** | **4.43** | **3.84** | **3.75** | **3.70** |

space character. The function COUNT maps each $W \in \mathbf{W}$ to its frequency in the corpus in UTF-8 format. The candidate token set $\mathbf{T}$ for GREEDTOK includes all substrings of words in $\mathbf{W}$ while UNIGRAM's is at the character level. In contrast, BPE begins with an empty $\mathbf{T}$ and builds tokens incrementally from $\mathbf{B}$, depending on the final $|\mathbf{S}| = k$. Thus, both GREEDTOK and BPE produce final token sets of size $|\mathbf{B}| + k$. However, in addition, we allow UNIGRAM to include frequent (possibly multibyte) characters in the final vocabulary due to the sentencepiece implementation, i.e. it has a final token set size larger than $|\mathbf{B}| + k$.

**Discussion.** We see from Table 2 that GREEDTOK consistently uses fewer tokens on average to represent the same data. Since BPE relies on repeated applications of merge rules to build large tokens, this suggests that many intermediate tokens created during merging may never be used in the final encoding, effectively wasting vocabulary capacity that could be allocated to more useful tokens. Meanwhile, we know from our example in Appendix D that UNIGRAM can over-prioritize whole words at the expense of informative subword tokens, and the empirical results of Table 2 confirms our suspicion that such suboptimal scenarios are not rare.

Table 1: Dataset statistics and the time taken for compute with word counts as inputs, conducted with AMD EPYC 9654 @ 2.40GHz. Refer to Appendix F.1 for additional dataset descriptions.

| Dataset | $|\mathbf{W}|$ | $\sum_W^{\mathbf{W}} c_W$ | $|\mathbf{T}|$ | max $|\mathbf{S}|$ | Time |
|---|---|---|---|---|---|
| UN | 105K | 37M | 884K | 5K | 6s |
| arχiv | 881K | 366M | 7,626K | 5K | 63s |
| wiki | 8,769K | 2,949M | 93.5M | 10K | 11m |
| PubMed | 6,527K | 4,149M | 121M | 10K | 11m |

### 5.2 Evaluating GREEDTOK's language pre-training

In Appendix E, we show empirical evidence that the token sets produced by GREEDTOK more closely resemble UNIGRAM than BPE, suggesting that they may inherit some of UNIGRAM's favorable token characteristics. To test this hypothesis, we pretrain two 1B-parameter language models (details in Appendix F.2), differing only in tokenizer choice — GREEDTOK versus BPE.[4] Both models use a vocabulary size of 65,536 and are trained on approximately 20% of the DCLM Dedup dataset [TGD+24, LFS+24], with their final token sets being 75% similar.

Table 3: The token count statistics for all three settings. GREEDTOK uses nearly 18% fewer tokens to represent the entire DCLM Dedup dataset. The total training tokens used is around 629B tokens.

| Experiment Name | Tokenizer | Full dataset tokens | Training tokens | Dataset % |
|---|---|---|---|---|
| BPEM | BPE | $8.94 \cdot 10^{11}$ | $6.29 \cdot 10^{11}$ | 70.35% |
| Equal Tokens (GTET) | GREEDTOK | $7.35 \cdot 10^{11}$ | $6.29 \cdot 10^{11}$ | 85.58% |
| Equal Proportion (GTEP) | GREEDTOK | $7.35 \cdot 10^{11}$ | $5.03 \cdot 10^{11}$ | 68.47% |

---

[4]We use BPE as the baseline, given its status as the most widely adopted tokenizer for LLMs [KHM+23].

We compare the BPE-based model (BPEM) against two versions (GTET and GTEP) of the GREEDTOK-based model under different training constraints, summarized in Table 3:

1. **GREEDTOK Equal Tokens (GTET).** Trained using the same number of tokens as BPEM. This setting isolates the impact of denser token representations by fixing the token count.

2. **GREEDTOK Equal Proportion (GTEP).** Trained using the same proportion of the original dataset as BPEM. Here, the number of training tokens differs, based on each tokenizer's compression ratio, allowing us to examine the effect of using fewer tokens from equivalent text coverage.

**Evaluation.** We use the popular `Language Model Evaluation Harness` [GTA+24] toolkit and their predefined evaluation settings to evaluate BPEM, GTET, and GTEP. Several popular benchmark sets were used for evaluation, refer to Appendix F.3 for more information.

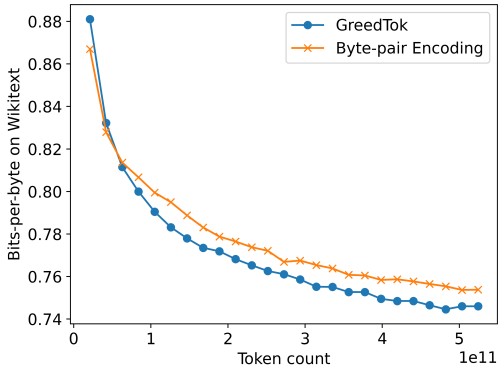

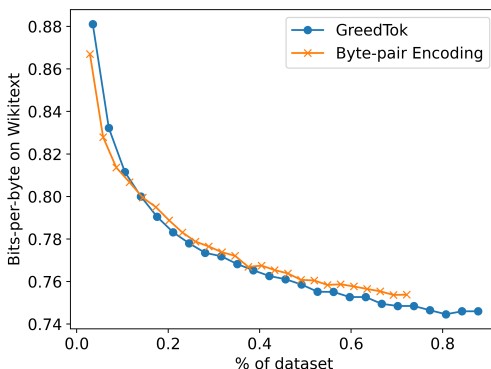

(a) Comparing along number of tokens trained.

(b) Comparing along amount of text trained.

Figure 2: We plot the bits/byte improvement across phase 1 training for model using GREEDTOK and BPE on different scales. The bits/byte metric is independent of tokenization and reflects true compression performance on the underlying data. Since both GTET and GTEP are equivalent in phase 1 for the first 100,000 steps, we examine bits/byte improvement on Wikitext with different scales on the x-axes.

Table 4: Evaluation results on popular benchmarks. GTET/GTEP obtained better scores than BPEM.

| | Accuracy (normalized) | | | | | | Accuracy | | | | | | bits/byte |
|---|---|---|---|---|---|---|---|---|---|---|---|---|---|
| | ARC-c | ARC-e | Hella-Swag | OBQA | PIQA | SciQ | BoolQ | COPA | LAMB-BADA | RACE | Wino-grande | Avg. | Wikitext |
| BPEM | 36.2 | 67.9 | 65.6 | 40.0 | 75.7 | 89.8 | 65.8 | 81.0 | 61.1 | 36.4 | 62.8 | 62.0 | 0.7066 |
| GTEP | 37.6 | 68.8 | 64.9 | 39.6 | 75.6 | 90.0 | 67.6 | 79.0 | 63.9 | 36.8 | **63.5** | 62.5 | 0.7028 |
| GTET | **38.3** | **70.0** | **65.7** | **40.6** | **75.8** | **90.5** | **67.7** | **82.0** | **64.6** | **37.7** | 62.6 | **63.2** | **0.6989** |

**Discussion.** Previous works report either similar or a decrease in performance stemming from better compression [Gal19, GCE+24, SRZ+24, AFT+24]. One plausible explanation for these results is that, given any sentence, a tokenizer with a lower compression rate uses more tokens, which results in a higher number of total activations in a transformer during inference. This increase in the effective *width* of the transformer's computation circuit can increase its expressive power [PMB24], resulting in better performance for poorer compression. However, the results in Table 4 show that GREEDTOK, compared to its BPE counterpart, has achieved better compression while still maintaining model performance. This suggests an example of meaningful compression, with both GTET and GTEP outperforming BPEM on the evaluated benchmarks. From Fig. 2a, when training on equal token count, GREEDTOK is ahead. While Fig. 2b shows that, when normalized and trained on equivalent byte-lengths of data, GREEDTOK performs comparably to BPE highlighting the competitive modeling capacity of GREEDTOK, despite structural differences in tokenization. Our results suggest that the higher compression rate of GREEDTOK does not negatively affect downstream performance. Furthermore, it is even possible to achieve the same results with GREEDTOK while using fewer tokens for training.

## 5.3 Towards understanding GREEDTOK's approximability

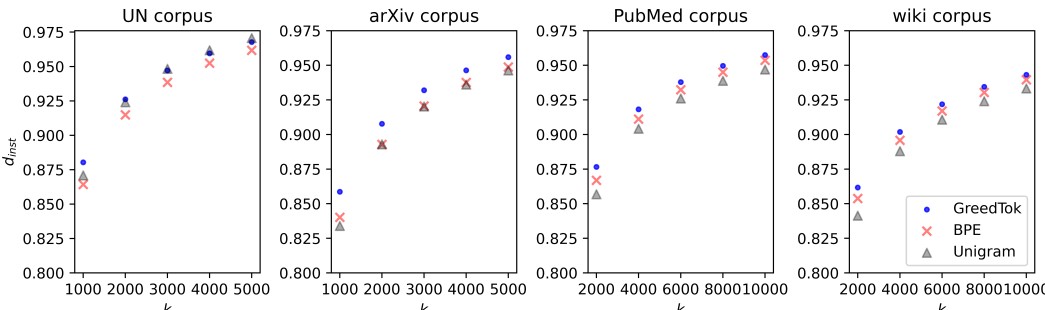

Figure 3: Plots showing exact $d_{\text{inst}}$ of each problem instance at different $|\mathbf{S}| = k$. As $k$ increases, the ratio of objectives $d_{\text{inst}}$ between GREEDTOK/BPE/UNIGRAM and GREEDWMC closes to 1.

We reformulated Problem 1 into a MIP in Section 4.1 as it relaxes naturally into the maximum coverage problem, which has a corresponding $(1 - 1/e)$ approximate algorithm GREEDWMC. Using GREEDWMC on the same problem instances of similar $k$, we can calculate the ratio of objectives between GREEDTOK and GREEDWMC and define $d_{\text{inst}} = \frac{\text{GREEDTOK}}{\text{GREEDWMC}}$ for each instance. Therefore, GREEDTOK attains an objective value *at least* $d_{\text{inst}}(1 - 1/e)$ times the optimal objective of Eq. (2) by definition, with GREEDWMC's objective value is *at least* that of TOK; see Section 4.2.

**Discussion.** From Fig. 3, for the four selected corpora, we plot $d_{\text{inst}}$ against $k$, observing that as $k$ increases, $d_{\text{inst}}$ climbs towards 1. In addition, to ensure that the results are generalizable to the wider internet corpus, we evaluate on `RefinedWeb` corpus, and obtain similar findings; see Fig. 4. Empirically, GREEDTOK achieves an objective value of at least $0.9(1 - 1/e)$ of the optimal for large $k$, relevant for practical NLP scenarios. With $d_{\text{inst}} \to 1$, this implies that the room for possible compression improvements narrows.

## 6 Conclusion

In this work, we showed that the tokenization problem is NP-hard and provided a greedy algorithm GREEDTOK that is a practical alternative over incumbents BPE and UNIGRAM, and may even be a better option for language pre-training. Although recent works [HKM+24, GT24] had pushed the limits of context length, plausibly reducing the importance of compression, GREEDTOK can still offer a flexible platform to explore new alternate objectives, such as integrat-

Figure 4: We sampled documents from `RefinedWeb` corpus at a probability of 0.01 across 40 independent runs, then run GREEDTOK. Plotting mean $d_{\text{inst}}$ shows it trending towards 1, empirically showing GREEDTOK is a $0.9(1 - 1/e)$-approximate algorithm.

ing NLP downstream objectives [BD20] and fairness [LBG+24] constraints into its MIP formulation. Finally, recall that the tokenization problem has the confounding property of being neither supermodular nor submodular. Although we show that GREEDTOK achieves an approximation ratio of at least $0.9(1 - 1/e)$ for large $k$, a formal proof is lacking. Nevertheless, this is an intriguing theoretical problem. We hope that our formulation of the tokenization problem and the accompanying toolkit will be valuable for future research.

**Computational feasibility.** While the theoretical runtime of GREEDTOK for selecting $\mathbf{S}$ is $O(|\mathbf{T}| \cdot k \cdot \sum_{W \in \mathbf{W}} |W|)$, a key optimization is to update a token $T$'s marginal contribution only when it is being evaluated for inclusion. Empirically, we observe that this lazy evaluation strategy scales like $\Theta(|\mathbf{T}| \cdot \sum_{W \in \mathbf{W}} |W|)$, making GREEDTOK practical for large-scale NLP workloads. Table 1

summarizes the time for GREEDTOK to compute $\mathbf{S}$ at the largest tested size $|\mathbf{S}| = \max k$ (see Section 5.1). In a larger experiment, with $|\mathbf{W}| = 14.3M$ and $|\mathbf{T}| = 251M$, GREEDTOK computed $\mathbf{S}$ in 34 minutes using 160GB of RAM. This cost can be reduced by reducing the search space, limiting $\max |W|$, filtering $\mathbf{W}$, or pruning $\mathbf{T}$ by substring length or frequency. To benchmark encoding speed, we tokenize a subset of the `wiki` corpus (70K articles, 97M words) using a vocabulary of $|\mathbf{S}| = 100K$ (from `cl100k_base` in TIKTOKEN [Ope23]). Our current implementation of GREEDTOK achieves 700K–800K words per second per thread, and we expect further optimization is possible. These results demonstrate that GREEDTOK is feasible for integration into modern NLP pipelines.

**Future extensions.** There are many tokenization techniques that augment an initial token set produced from core tokenization algorithms like BPE and UNIGRAM. Likewise, these methods could also be used to augment the token sets produced from GREEDTOK. For example, BPE-DROPOUT [PEV20] introduces stochasticity by randomly dropping merge operations during training, yielding multiple possible segmentations per input. While GreedTok is deterministic by default, it can be adapted in a similar fashion: at each step, we can randomly skip adding the top token and proceed with updating the graph accordingly. PATHPIECE [SRZ+24], an encoding algorithm, can be applied directly to any token set, including the ones generated by GreedTok. PICKYBPE [CAKY24] refines BPE vocabulary by iteratively removing low-utility tokens, using a deletion criterion guided by a hyperparameter. For encoding, it relies on a naive greedy approach or PATHPIECE. GREEDTOK can likewise incorporate such retrospective pruning: after each token addition, evaluate and remove earlier redundant tokens to improve vocabulary efficiency. BOUNDLESSBPE [SRTP25] and SUPERBPE [LHH+25] are contemporaneous methods that allow token merging across whitespace boundaries by generating longer tokens from an initial BPE token set resulting in larger token sets. GREEDTOK tokens could be used as an initial set from which these whitespace-spanning merges are constructed. VOLT [XZG+21] prunes an initial token set, generated via Unigram or BPE, by seeking to maximize the entropy of subword distributions. Again, GREEDTOK could serve as an upstream tokenizer to generate the initial candidate tokens for VOLT.

**Limitations.** The purpose of this work is to offer a new perspective on tokenization, with empirical experiments to show that this theory is practical. Pretraining language models from scratch is expensive, hence, our comparisons are limited to BPE, since it is widely adopted in current practice, and we fixed our models to have a size of 1B parameters. Although we believe that our insights and observed trends should generalize to larger models, more empirical confirmation at scale is needed. Our experiments mainly use corpora that contain commonly used languages, and did not conduct evaluations on low-resource languages, which is an important area for further exploration.

## Acknowledgements

This research/project is supported by the National Research Foundation, Singapore under its AI Singapore Programme (AISG Award No: AISG3-PhD-2023-08-055T).

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

# A  Example illustrating that Problem 1 is neither submodular nor supermodular

In this section, we show that the objective of Problem 1 is neither submodular nor supermodular. Using $2^{\mathbf{T}}$ to denote the powerset of $\mathbf{T}$, submodular and supermodular set functions are defined as follows:

**Definition 2** (Submodular Set Function). A real-valued set function $f : 2^{\mathbf{T}} \to \mathbb{R}$ is submodular if $f(\mathbf{A} \cup \{C\}) - f(\mathbf{A}) \geq f(\mathcal{A} \cup \{C\}) - f(\mathcal{A})$ for all $\mathbf{A} \subseteq \mathcal{A} \subseteq \mathbf{T}$ and $C \in \mathbf{T} \setminus \mathcal{A}$.

**Definition 3** (Supermodular Set Function). A real-valued set function $f : 2^{\mathbf{T}} \to \mathbb{R}$ is supermodular if $f(\mathbf{A} \cup \{C\}) - f(\mathbf{A}) \leq f(\mathcal{A} \cup \{C\}) - f(\mathcal{A})$ for all $\mathbf{A} \subseteq \mathcal{A} \subseteq \mathbf{T}$ and $C \in \mathbf{T} \setminus \mathcal{A}$.

In the context of the tokenization problem, the set $\mathbf{T}$ represents the candidate set of all possible tokens. Unfortunately for us, Problem 1 is neither submodular nor supermodular; see Table 5.

Table 5: The above table shows that the objective function of Problem 1 is neither supermodular nor submodular. Suppose we wish to tokenize the word $W = $ scaredy with candidate token set $\mathbf{T} = \{$care, edy, scar, scared, dy$\}$ and singletons $\{$s,c,a,r,e,d,y$\}$, and the function $f$ outputs the *smallest possible number of final tokens* used to represent $W$, i.e. the objective function of Problem 1 on a single word corpus. Observe that $\mathbf{X} \subseteq \mathbf{Y} \subseteq \mathbf{T}$, $Z \in \mathbf{T} \setminus \mathbf{Y}$, $\mathbf{X} \subseteq \mathbf{Y}' \subseteq \mathbf{T}$, and $Z' \in \mathbf{T} \setminus \mathbf{Y}'$. In case 1, using $\mathbf{X}$ to tokenize $W$ results in using 4 tokens (s, care, d, y) and one can check that using $\mathbf{X} \cup \{Z\}$ also results in 4 tokens. On the other hand, using $\mathbf{Y}$ results in 4 tokens (s, care, d, y) but using $\mathbf{Y} \cup \{Z\}$ results in 2 tokens (scar, edy). Therefore, $f(\mathbf{X} \cup \{Z\}) - f(\mathbf{X}) > f(\mathbf{Y} \cup \{Z\}) - f(\mathbf{Y})$ and thus $f$ is *not* supermodular. On the other hand, in case 2, using $\mathbf{Y}'$ and $\mathbf{Y}' \cup \{Z'\}$ to tokenize $W$ results in 2 tokens (scared, y) while using $\mathbf{X} \cup \{Z'\}$ results in 3 tokens (s, care, dy). Therefore, $f(\mathbf{X} \cup \{Z'\}) - f(\mathbf{X}) < f(\mathbf{Y}' \cup \{Z'\}) - f(\mathbf{Y}')$ and thus $f$ is *not* submodular.

| Single word corpus $W = $ scaredy with COUNT$(W) = 1$ | | | |
|---|---|---|---|
| Case 1 | | Case 2 | |
| $\mathbf{X} = \{$care$\}$ $\quad$ $f(\mathbf{X}) = f(\mathbf{X} \cup \{Z\}) = 4$ | | $\mathbf{X} = \{$care$\}$ $\quad$ $f(\mathbf{X}) = 4$ | |
| $\mathbf{Y} = \{$care, edy$\}$ $\quad$ $f(\mathbf{Y}) = 4$ | | $\mathbf{Y}' = \{$care, scared$\}$ $\quad$ $f(\mathbf{Y}') = f(\mathbf{Y}' \cup \{Z'\}) = 2$ | |
| $Z = $ scar $\quad$ $f(\mathbf{Y} \cup \{Z\}) = 2$ | | $Z' = $ dy $\quad$ $f(\mathbf{X} \cup \{Z'\}) = 3$ | |
| $0 = f(\mathbf{X} \cup \{Z\}) - f(\mathbf{X}) > f(\mathbf{Y} \cup \{Z\}) - f(\mathbf{Y}) = -2$ | | $-1 = f(\mathbf{X} \cup \{Z'\}) - f(\mathbf{X}) < f(\mathbf{Y}' \cup \{Z'\}) - f(\mathbf{Y}') = 0$ | |

# B  Mixed integer program formulation

In this section, we give full details of our mixed integer program (MIP) formulation and provide examples for better understanding.

To formulate Problem 1 as an MIP, our goal is to choose a subset $\mathbf{S} \subseteq \mathbf{T}$ of size $|\mathbf{S}| \leq k$ such that the following objective is maximized, encoding $\max \sum_{W \in \mathbf{W}} \text{COUNT}(W) \cdot \text{COVER}(W, \mathbf{S})$:

$$\max \sum_{W \in \mathbf{W}} c_W \cdot \left( \sum_{i=1}^{|W|-1} m_{i,i+1}^{W} \right) \tag{2}$$

with the following binary variables, where $c_W = \text{COUNT}(W)$:

- $x_T \in \{0, 1\}$, for all tokens $T \in \mathbf{T}$
  *Did we choose token $T \in \mathbf{T}$, i.e. $T \in \mathbf{S}$?*

- $m_{1,2}^{W}, \ldots, m_{|W|-1,|W|}^{W} \in \{0, 1\}$, for all words $W \in \mathbf{W}$
  *Are the $i^{th}$ singleton $W_i$ and the $(i+1)^{th}$ singleton $W_{i+1}$ covered by the same token?*

- $m_{1,2}^{W,T}, \ldots, m_{|W|-1,|W|}^{W,T} \in \{0, 1\}$, for all words $W \in \mathbf{W}$ and tokens $T \in \mathbf{T}$
  *Did token $T \in \mathbf{S}$ cover the $i^{th}$ singleton $W_i$ and the $(i+1)^{th}$ singleton $W_{i+1}$?*

under the following constraints:

$$\sum_{T \in \mathbf{T}} x_T \leq k \tag{3}$$

$$x_T \geq m_{i,i+1}^{W,T} \qquad \text{if } (W_i, W_{i+1}) \subseteq T \qquad \forall W \in \mathbf{W}, \forall T \in \mathbf{T}, \forall i \in \{1, \dots, |W|-1\} \tag{4}$$

$$\sum_{T \in \mathbf{T}} m_{i,i+1}^{W,T} \geq m_{i,i+1}^{W} \qquad \forall W \in \mathbf{W}, \forall T \in \mathbf{T}, \forall i \in \{1, \dots, |W|-1\} \tag{5}$$

$$\sum_{T \in \mathbf{T}} m_{i,i+1}^{W,T} \leq 1 \qquad \forall W \in \mathbf{W}, \forall T \in \mathbf{T}, \forall i \in \{1, \dots, |W|-1\} \tag{6}$$

$$m_{i,i+1}^{W,T} = m_{i+1,i+2}^{W,T} \qquad \text{if } (W_i, W_{i+1}, W_{i+2}) \subseteq T \qquad \forall W \in \mathbf{W}, \forall T \in \mathbf{T}, \forall i \in \{1, \dots, |W|-2\} \tag{7}$$

$$\sum_{T \in \mathbf{T}} m_{s-1,s}^{W,T} \leq 1 - m_{s,s+1}^{W,T} \qquad \text{if } T \text{ starts with } (W_s, W_{s+1}) \quad \forall W \in \mathbf{W}, \forall T \in \mathbf{T}, \forall s \in \{2, \dots, |W|-1\} \tag{8}$$

$$\sum_{T \in \mathbf{T}} m_{e,e+1}^{W,T} \leq 1 - m_{e-1,e}^{W,T} \qquad \text{if } T \text{ ends with } (W_{e-1}, W_e) \quad \forall W \in \mathbf{W}, \forall T \in \mathbf{T}, \forall e \in \{2, \dots, |W|-1\} \tag{9}$$

We remark that the objective Eq. (2) can be re-expressed as $\max \sum_{T \in \mathbf{T}} \sum_{W \in \mathbf{W}} \sum_{i=1}^{|W|-1} c_W \cdot m_{i,i+1}^{W,T}$, making Eq. (5) redundant. However, this current formulation is useful for showing how to relax TOK to WMC later.

Now, let us interpret and explain the constraints. Eq. (3) models the constraint that we are choosing a subset of size $|\mathbf{S}| \leq k$. Eq. (4) models the constraint that we can only use $T \in \mathbf{T}$ to cover if it is chosen in $\mathbf{S}$. Eq. (5) models the constraint that if a cover happened between two adjacent singletons, then a relevant $T \in \mathbf{T}$ must have been chosen in $\mathbf{S}$. However, Eq. (6) models the constraint of only covering two adjacent singletons with a single relevant $T \in \mathbf{S}$. Eq. (7) models the constraint of covering the entire substring $T \in \mathbf{T}$, or leave it uncovered. Eq. (8) and Eq. (9) model the constraints preventing the chosen substring $T \in \mathbf{T}$ from sharing the cover with another partially overlapping $T$.

In the following examples, we succinctly write $m^W$ and $m^{W,T}$ in the forms of $(m_{1,2}^W, m_{2,3}^W, \dots, m_{|W|-1,|W|}^W)$ and $(m_{1,2}^{W,T}, m_{2,3}^{W,T}, \dots, m_{|W|-1,|W|}^{W,T})$ respectively, for any word $W \in \mathbf{W}$ and token $T \in \mathbf{T}$.

**Example 2.** Consider the word $W = \text{ababa}$ and the token $T = \text{aba}$ has $x_T = 1$, i.e. $T \in \mathbf{S} \subseteq \mathbf{T}$. If we *only use* $T$ to cover singletons in $W$ with left-to-right priority, then the resultant tokenized form of $W$ is aba␣b␣a. So, $m^W = (1,1,0,0)$, $m^{W,T} = (1,1,0,0)$, and $m^{W,T'} = (0,0,0,0)$ for all $T' \in \mathbf{T} \setminus \{T\}$. Observe that the 0 bits in $m^W$ precisely denote the partitioning positions within $W$. Furthermore, the constraints Eq. (6) and Eq. (7) ensure that $T$ is the only token that occupies the first two adjacent singletons, while constraints Eq. (8) and Eq. (9) prevent an invalid overlap of $T$ for the last two adjacent singletons. Now, suppose if we also have $T' = \text{ba}$ with $x_{T'} = 1$. Using both $T$ and $T'$ to tokenize $W$ results in aba␣ba with $m^W = (1,1,0,1)$, $m^{W,T} = (1,1,0,0)$, $m^{W,T'} = (0,0,0,1)$, and $m^{W,T''} = (0,0,0,0)$ for all $T'' \in \mathbf{T} \setminus \{T, T'\}$.

**Example 3.** Tokenizing the word $W = \text{abcdef}$ using only tokens $S_1 = \text{bc}$ and $S_2 = \text{de}$ yields a␣bc␣de␣f. This corresponds to $m^W = (0,1,0,1,0)$, $m^{W,S_1} = (0,1,0,0,0)$, $m^{W,S_2} = (0,0,0,1,0)$, and $m^{W,T} = (0,0,0,0,0)$ for all $T \in \mathbf{T} \setminus \{S_1, S_2\}$. Meanwhile, tokenizing the word $W = \text{abcdef}$ using only token $S_3 = \text{bcde}$ yields a␣bcde␣f, corresponding to $m^W = (0,1,1,1,0)$, $m^{W,S_3} = (0,1,1,1,0)$, and $m^{W,T} = (0,0,0,0,0)$ for all $T \in \mathbf{T} \setminus \{S_3\}$. Observe that using token $S_3$ alone directly accomplishes what a typical bottom-up pairwise merge sequence from BPE would do: apply $S_1$ to merge 'b' with 'c', $S_2$ to merge 'd' with 'e', then $S_3$ to merge 'bc' with 'de'.

## C Relation to the weighted maximum coverage problem

In this section, we provide details on how our MIP formulation in Section 4.1 naturally relaxes into the well known weighted maximum coverage problem (WMC).

Given a set of unique elements $\mathbf{L} = \{L_1, \ldots, L_{|\mathbf{L}|}\}$ and their corresponding weights $\mathcal{W} = \{w_1, \ldots, w_{|\mathbf{L}|}\}$, a collection of sets $\mathbf{U} = \{U_1, \ldots, U_{|\mathbf{U}|}\}$ where $U \in \mathbf{U} \subseteq \mathbf{L}$, and a number $k$, we want to find a subset $\mathbf{U}' \subseteq \mathbf{U}$ such that $|\mathbf{U}'| \leq k$ and the total weights of covered elements $\sum_{L_i \in \bigcup \mathbf{U}'} w_i$ is maximized. Formulating WMC as a mixed integer program, we have the objective:

$$\max \sum_{L_i \in \mathbf{L}} w_i \ell_i \tag{10}$$

with the following variables:

- $\ell_i \in \{0, 1\}$, for all $L_i \in \mathbf{L}$
  *Did we choose element $L_i \in \mathbf{L}$, i.e. is $L_i$ covered?*
- $\mu_j \in \{0, 1\}$, for all $U_j \in \mathbf{U}$
  *Did we choose set $U_j \in \mathbf{U}$, i.e. is $U_j \in \mathbf{U}'$?*

under the following constraints:

$$\sum_{U_j \in \mathbf{U}} \mu_j \leq k \tag{11}$$

$$\sum_{L_i \in U_j}^{\mathbf{U}} \mu_j \geq \ell_i \qquad \forall \ell_i \in \mathbf{L} \tag{12}$$

Let us now interpret and explain the constraints. Eq. (11) limits the number of selected sets $\leq k$. Eq. (12) ensures that if an element is covered, at least one of the sets containing the element must be included in $\mathbf{U}'$. To see that WMC is a relaxation of TOK, we first establish a mapping between the variables between TOK and WMC:

- $m_{i,i+1}^{W} \to \ell_i$
  *decision of covering adjacent singletons $\to$ decision of covering element*
- $x_T \to \mu_j$
  *decision of including $T \in \mathbf{S} \to$ decision of including $U \in \mathbf{U}'$*
- $m_{i,i+1}^{W,T} \to L_i \in U_j$
  *adjacent singletons in $W$ and $T \to$ element membership in set*
- $\sum_{W \in \mathbf{W}} c_W \to w_i$
  *sum count of $W$ with adjacent singletons $\to$ weight of element*

Next, comparing the objectives, we can see that Eq. (2) and Eq. (10) have the exact same objective when utilizing the mapping between variables. Finally, we demonstrate a relaxation of TOK's constraints:

- Eq. (3) and Eq. (11) are equivalent
  *select at most $k$ number of $T$ and $U$ respectively*
- Combining Eq. (4) and Eq. (5) gives us Eq. (12).
  $\sum_{T \in \mathbf{T}} x_T \geq \sum_{T \in \mathbf{T}} m_{i,i+1}^{W,T} \geq m_{i,i+1}^{W} \to \sum_{L_i \in U_j}^{\mathbf{U}} \mu_j \geq \ell_i$
- We remove the constraints Eq. (6), Eq. (7), Eq. (8), and Eq. (9).

Notice that for TOK, we disentangle Eq. (4) and Eq. (5) using the specification of $m_{i,i+1}^{W,T}$ to enable Eq. (6), limiting the covering of an element to one selected set.

# D TOK's relation to UNIGRAM

Let $\mathbf{T}$ represent the set of all possible subword sequences. The probability of a subword sequence $\vec{W} = (T_1, \ldots, T_M)$ where $T \in \mathbf{T}$ formulated as a product of subword probabilities:

$$P(\vec{W}) = \Pi_{i=1}^M p(T_i)$$

where $\sum_T^{\mathbf{T}} p(T) = 1$. Since a word $W$ in corpus $D$ can be represented by different possible subword sequences $\mathcal{S}(W)$, let $W^*$ be the most probable segmentation:

$$W^* = \underset{\vec{W} \in \mathcal{S}(W)}{\operatorname{argmax}} P(\vec{W})$$

Since $\mathcal{S}(D_s)$, segmentation candidates of sentence $D_s$, will be individual words $w$ (based on `sentencepiece` default implementation). Therefore, Unigram seeks to minimize the reduction in likelihood $\mathcal{L}$ amongst words in given corpus:

$$\max \mathcal{L} = \sum_{s=1}^{|D|} \sum_{W \in \mathcal{S}(D_s)} \log W^*$$

To map $\mathcal{L}$ to TOK's objective of $\min \sum_W^{\mathbf{W}} \text{COUNT}(W) \cdot partition(W)$:

$$
\begin{aligned}
\max \mathcal{L} &= \sum_{s=1}^{|D|} \sum_{w \in \mathcal{S}(D_s)} \log W^* \\
&= \sum_W^{\mathbf{W}} \text{COUNT}(W) \cdot \log W^* & \text{(group by words)} \\
&= \sum_W^{\mathbf{W}} \text{COUNT}(W) \cdot \log \Pi_{W_i}^{W^*} p(W_i) & \text{(choose best subword segmentation)} \\
&= \sum_W^{\mathbf{W}} \text{COUNT}(W) \cdot (\log p(W_i^*) + \cdots + \log p(W_{|W^*|}^*)) & \text{(notice } partition(W) = |W^*|)
\end{aligned}
$$

Notice that after grouping all similar words together, we get a weighted partition $(\log p(W_i) + \cdots + \log p(W_{|W^*|}))$. For TOK, it is $(1 + \cdots + 1) = |W^*|$. Maximizing $\mathcal{L}$ is equivalent to minimizing negative log probability weighted partitions. Due to the presence of $\log$, $\max \mathcal{L}$ is equivalent to $\min$ weighted TOK. We can see that Unigram favors frequently occurring subwords in a non-linear fashion. This relation is also noted in [SRZ+24], where their proposed PATHPIECE tokenization algorithm optimizes for TOK from top-down pruning of BPE/UNIGRAM shortlisted candidates.

We present a scenario where top-down pruning is sub-optimal for compression. Given $\mathbf{W} = \{\text{"random"}, \text{"randose"}, \text{"rosey"}, \text{"randy"}\}$, each with a COUNT of 1, and $\mathbf{T} = \mathbf{W} \bigcup \{\text{"rand"}, \text{"ose"}\}$, we wish to select $\mathbf{S} \subset \mathbf{T}$, where $|\mathbf{S}| = k = 2$ tokens.

Table 6 shows the token pruning process of UNIGRAM and Table 7 shows the greedy approach of GREEDTOK. For this scenario, GREEDTOK obtained a better solution with 10 partitions compared to UNIGRAM's 12. This scenario highlights the skew to select whole words when approaching tokenization from a pruning angle. If such scenarios are common, then we can expect GREEDTOK to obtain a better tokens per word ratio in our experiments (Section 5.1).

Table 6: Top-down pruning solution for the given example using UNIGRAM, removing tokens that results in the least decrease in $\mathcal{L}$, $\triangle\mathcal{L}$, in each iteration. The same results will also be obtained when optimizing for TOK from a top-down pruning approach. For iteration 2, $b$ denotes the singletons which are omitted. Final $\mathbf{S} = \{$"random", "randose"$\}$.

| | **W** | $\log W*$ | **T** | COUNT | $p(T)$ | | Removing results in new segments | $\triangle\mathcal{L}$ |
|---|---|---|---|---|---|---|---|---|
| **Iteration 1** | random | -1.505 | random | 1 | 0.0312 | random | random* = $p($"rand"$) \cdot p($"o"$) \cdot p($"m"$)$ | -2.056 |
| | randose | -1.505 | randose | 1 | 0.0312 | randose | randose*= $p($"rand"$) \cdot p($"ose"$)$ | -0.727 |
| | rosey | -1.505 | rosey | 1 | 0.0312 | rosey | randy* = $p($'r'$) \cdot p($"ose"$) \cdot p($'y'$)$ | -1.806 |
| | randy | -1.505 | randy | 1 | 0.0312 | randy | rosye* = $p($"rand"$) \cdot p($'y'$)$ | -0.727 |
| | | | **rand** | 3 | 0.0937 | rand | None | **0** |
| | | | **ose** | 2 | 0.0625 | ose | None | **0** |
| | Decision after iteration 1: remove "rand" and "ose", next iteration: | | | | | | | |
| **Iteration 2** | random | -1.431 | random | 1 | 0.0370 | random | random* = $\sum_{b\in W^*} p(b)$ | -4.646 |
| | randose | -1.431 | randose | 1 | 0.0370 | randose | random* = $\sum_{b\in W^*} p(b)$ | -5.475 |
| | rosey | -1.431 | **rosey** | 1 | 0.0370 | rosey | rosey* = $\sum_{b\in W^*} p(b)$ | **-3.743** |
| | randy | -1.431 | **randy** | 1 | 0.0370 | randy | randy* = $\sum_{b\in W^*} p(b)$ | **-3.391** |
| | Decision after iteration 2: remove "rosey" and "randy". Total partitions = 1 + 1 + 5 + 5 = **12**. | | | | | | | |

Table 7: GREEDTOK's solution for the given example, selecting tokens that results in the highest objective gain in each iteration. Final $\mathbf{S} = \{$"rand", "rosey"$\}$ or $\{$"rand", "ose"$\}$.

| | | **Iteration 1** | | **Iteration 2** | |
|---|---|---|---|---|---|
| **W** | | **T** | obj. gain | **T** | obj. gain |
| random | | random | 5 | random | 2 |
| randose | | randose | 6 | randose | 3 |
| randy | | rosey | 4 | **rosey** | **4** |
| rosey | | randy | 4 | randy | 1 |
| | | ose | 4 | **ose** | **4** |
| | | **rand** | 9 | | |
| Decision | | pick "rand" | | pick "rosey" or "ose" | |
| if pick "rosey": Total partitions = 3 + 4 + 1 + 2 = **10** | | | | | |
| if pick "ose": Total partitions = 3 + 2 + 3 + 2 = **10** | | | | | |

# E  Analyzing GREEDTOK's Characteristics

After obtaining the token sets of GREEDTOK/BPE/UNIGRAM from our experiments (Section 5), along three dimensions: 1) proportion of common tokens in a pairwise comparison, 2) proportion of whole words and 3) UNIGRAM's $\mathcal{L}$ objective (Appendix D).

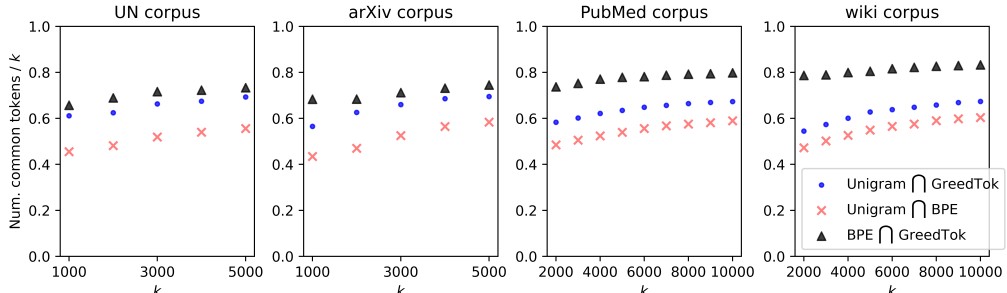

Figure 5: Plots showing the ratio of common tokens between token sets from GREED-TOK/BPE/UNIGRAM. From the plots, we observe that while GREEDTOK and BPE are most alike, GREEDTOK shares more similarties to UNIGRAM, compared to BPE.

**Common tokens.**  To probe GREEDTOK's token set, we analyze the common tokens from pairwise comparison of the tokenization methods investigated. From Fig. 5, we observe that BPE $\bigcap$ GREED-

TOK share a large proportion of common tokens. The remaining difference can be observed in the greater proportion of UNIGRAM ∩ GREEDTOK compared to UNIGRAM ∩ BPE. The consistent results across the four corpora, and at different $|\mathbf{S}| = k$, implies that GREEDTOK contains the characteristics of BPE due to the high proportionality of common tokens. However, we require further investigation into whether GREEDTOK contains UNIGRAM's characteristics.

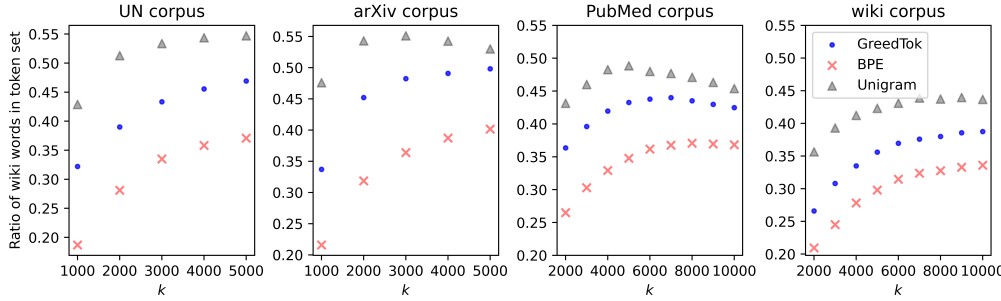

Figure 6: Plots showing the ratio of wiki words found in the token sets from GREED-TOK/BPE/UNIGRAM with different $|\mathbf{S}| = k$. We select the top 40K frequently occuring words excluding stop words to approximate a frequent word list. From the plots, UNIGRAM has the largest ratio of wiki words, followed by GREEDTOK and then BPE.

**Whole words.**   As hypothesized in Appendix D, there are scenarios that will lead UNIGRAM to select whole words, skipping the intermediate tokens that BPE may require. To further investigate, we approximate a 40,0000 common word list using the most frequent words appearing in `wiki`, and then use it to compare to the token sets obtained. From Fig. 6, across the four corpora and at different $|\mathbf{S}| = k$, we observe that UNIGRAM has the highest proportion of whole words, then followed by GREEDTOK and BPE respectively. With GREEDTOK selecting more whole words, compared to BPE, suggests that GREEDTOK may also exhibit the behaviour favored by UNIGRAM.

Table 8: In this table, we report $\mathcal{L}$ obtained by GREEDTOK/BPE/UNIGRAM under various $|\mathbf{S}| = k$ settings. We observe that GREEDTOK, compared to BPE, achieving a closer $\mathcal{L}$ to UNIGRAM.

| | k | 1000 | 2000 | 3000 | 4000 | 5000 | | 2000 | 4000 | 6000 | 8000 | 10000 |
|---|---|---|---|---|---|---|---|---|---|---|---|---|
| GREEDTOK (GT) $\mathcal{L}$ | | -4.15E08 | -3.85E08 | -3.69E08 | -3.60E08 | -3.53E08 | | -5.03E10 | -4.71E10 | -4.56E10 | -4.47E10 | -4.41E10 |
| BPE $\mathcal{L}$ | UN | -4.37E08 | -4.01E08 | -3.83E08 | -3.71E08 | -3.63E08 | PubMed | -5.19E10 | -4.83E10 | -4.66E10 | -4.56E10 | -4.49E10 |
| GT's Improvement (%) | | **5.08%** | **4.00%** | **3.48%** | **3.06%** | **2.71%** | | **3.19%** | **2.61%** | **2.26%** | **2.01%** | **1.68%** |
| UNIGRAM $\mathcal{L}$ | | -3.90E08 | -3.60E08 | -3.47E08 | -3.38E08 | -3.33E08 | | -4.87E10 | -4.56E10 | -4.43E10 | -4.35E10 | -4.31E10 |
| GT's Improvement (%) | | -6.29% | -6.77% | -6.54% | -6.39% | -6.09% | | -3.30% | -3.15% | -2.81% | -2.58% | -2.46% |
| GREEDTOK (GT) $\mathcal{L}$ | | -4.49E09 | -4.13E09 | -3.96E09 | -3.85E09 | -3.78E09 | | -3.76E10 | -3.55E10 | -3.44E10 | -3.37E10 | -3.32E10 |
| BPE $\mathcal{L}$ | arxiv | -4.74E09 | -4.34E09 | -4.12E09 | -3.99E09 | -3.89E09 | wiki | -3.85E10 | -3.62E10 | -3.50E10 | -3.42E10 | -3.37E10 |
| GT's Improvement (%) | | **5.37%** | **4.91%** | **3.99%** | **3.37%** | **2.94%** | | **2.34%** | **2.03%** | **1.84%** | **1.56%** | **1.41%** |
| UNIGRAM $\mathcal{L}$ | | -4.32E09 | -3.97E09 | -3.82E09 | -3.74E09 | -3.68E09 | | -3.61E10 | -3.42E10 | -3.32E10 | -3.26E10 | -3.22E10 |
| GT's Improvement (%) | | -3.99% | -4.10% | -3.51% | -3.00% | -2.59% | | -4.13% | -3.70% | -3.57% | -3.35% | -3.12% |

**Unigram's objective.**   Finally, another way that we can investigate the closeness of UNIGRAM and GREEDTOK, is to examine the $\mathcal{L}$ of their token sets. From Table 8, we observe that GREEDTOK's $\mathcal{L}$ is much closer to UNIGRAM's $\mathcal{L}$, relative to BPE's.

**Conclusion of token set investigation.**   From the three analyses, there are two key findings. First, GREEDTOK's compression ability can be explained by selecting a large proportion of tokens that BPE selects, and further improving it by selecting what BPE could not select, i.e. some tokens selected by UNIGRAM. Second, there are indicators that GREEDTOK may display UNIGRAM's pre-training quality. This is observed from GREEDTOK having a higher proportion of common tokens and a closer $\mathcal{L}$ to UNIGRAM relative to BPE. We investigate and confirm this in Section 5.2, where we conduct language pre-training on similar models trained on BPE versus GREEDTOK tokens.

# F  Additional Information

## F.1  Additional Corpora Information

In this subsection, we describe the corpora used in our compression experiments and detail any preprocessing steps.

**United Nations General Debate Corpus (UN).**  UN [JBD17] is a collection of statements made by various member states during the General Debate on key geopolitical issues from 1946 to 2022. This corpus has a Creative Commons (CC) 0: Public Domain License.

**ar$\chi$iv.**  This corpus[5] is a collection of abstracts of scientific publications and preprints from the popular e-Print archive. This corpus has a CC0: Public Domain License.

**Wikipedia-English (wiki).**  An extensive collection of English articles on a wide range of things, concepts, and people. We extract [Att15] the text from the database dump.[6] We also conduct a performance ablation with articles containing Chinese, Japanese and Korean (CJK) languages. The texts belonging to these articles are under CC BY-SA 4.0 and GNU Free Documentation Licenses.

**PubMed Central Open Access (PubMed).**  Similar to ar$\chi$iv, PubMed[7] is a repository of publications and preprints, mainly related to health, medicine, biotechnology, and pharmaceuticals. We select the Non-Commercial Use Only subset grouped by: CC BY-NC, CC BY-NC-SA, and CC BY-NC-ND licenses. We preprocessed the text minimally, removing citations and headers.

**RefinedWeb.**  This corpus [PMH+23] is a filtered set of CommonCrawl [Com23] with stringent filtering and extensive deduplication. It was used to pretrain language models with 1B and 7B parameters. The corpus has a ODC-By 1.0 license.

**DCLM full-dedup.**  This corpus is built from DCLM [LFS+24] using a deduplication process that was used to build Zyda-2 [TGD+24]. This corpus is licensed under CC-by-4.

## F.2  Model Pre-training Information

Our pre-training corpus is DCLM full-deduped dataset. For model training, we use the Dolomite Engine [Mis24]. Our model architecture is a 40-layer Transformer [VSP+17], with embedding size 1536, MLP using SwiGLU activation [Sha20] with intermediate size of 4096, and GQA [ALTdJ+23] layers with 12 query heads and 4 pairs of KV-heads. We used a fixed context length of 4096 tokens and a batch size of $2^{22} \approx 4M$ tokens. We train GREEDTOK and BPE tokenizers on approximately 20% of the dataset, randomly selected. For BPE, we use the popular implementation from Huggingface's tokenizer API.[8] During training, we follow the same learning rate schedule as [SSM+24], training on the dataset in two phases: the Power Scheduler [SSM+24] in phase 1 and a learning rate with exponential decay in phase 2. However, our phases 1 and 2 use a similar data mixture, sampling the first 500M documents for phase 1, and the next 100M documents for phase 2, which always use 20% of the training iterations of phase 1. We run our experiments using NVIDIA H100 80GB HBM3 cluster, with 96 logical CPU count, training at a rate of $\sim$400B tokens/day.

Evaluating GTET is analogous to a setting where model training is compute-constrained, and users can use more text data for training. Conversely, evaluating GTEP is analogous to a setting where model training is data-constrained, and users have a limited amount of text for training. In the BPEM and GTET settings, we train the model for 125,000 and 25,000 iterations in phases 1 and 2 respectively. For GTEP, we take the model checkpoint of GTET at the 100,000th training iteration step, followed by an additional 20,000 training iterations in phase 2.

---

[5]Available at: kaggle.com/datasets/Cornell-University/arxiv.
[6]Available at: https://dumps.wikimedia.org/.
[7]Available at: https://pmc.ncbi.nlm.nih.gov/tools/openftlist/.
[8]Available at: https://github.com/huggingface/tokenizers.

## F.3 Additional Benchmark Information

We use the default settings of `Language Model Evaluation Harness` [GTA⁺24] for our benchmark evaluations.

**ARC-Easy (ARC-e).** A subset of easy questions from the `Abstraction and Reasoning Corpus (ARC)` dataset [CCE⁺18] contains 2,376 grade school level multiple-choice questions that were answered correctly with retrieval-based algorithm and a word co-occurrence algorithm. Each question has 4 answer options.

**ARC-Challenge (ARC-c).** Another subset of questions from the ARC dataset, the test set comprises 1,172 grade school level multiple-choice questions that were incorrectly answered with retrieval-based algorithm and a word co-occurrence algorithm. Similar to the easy subset, each question has 4 answer options.

**HellaSwag.** This challenge set [ZHB⁺19] evaluates sentence completion in a multiple-choice setting. Given 4 possible answers, the correct answer completes the given sentence best. The test set contains 10,003 question sets.

**OpenBook QA (OBQA).** This dataset [MCKS18] contains multiple-choice questions that require additional reasoning and knowledge in addition to the information included in the question and its 4 answer choices. We use the main set containing 500 test question sets.

**Physical Interaction: Question Answering (PIQA).** This dataset [BZB⁺20] contains binary-choice physical commonsense questions. Additional knowledge of how physical materials interact is required to successfully answer the questions. We use the validation set of 2,000 question sets.

**SciQ.** This dataset [JW17] contains multiple-choice science questions that were crowdsourced. Each question has 4 answer choices. We use its test subset of 1000 question sets.

**BoolQ.** This dataset [CLC⁺19] contains yes/no questions. Each question is accompanied by a corresponding text passage. We use the validation set of 3,270 triplets of question, passage, and answer.

**Choice of Plausible Alternatives (COPA).** This dataset [RBG11] seeks to evaluate commonsense reasoning, where given a premise and two plausible causes or effects, the correct answer is the option that is more plausible than the other. We use the validation set of 500 question sets.

**LAMBADA.** This dataset [PKL⁺16], abbreviated from `LAnguage Modeling Broadened to Account for Discourse Aspects`, contains text passages with the given task of predicting the last word of the target sentence. The task was crafted in a manner that the target word cannot be predicted by the target sentence, requiring information from other parts of the given text passage. We use its test set containing 5,153 text passages.

**ReAding Comprehension dataset from Examinations (RACE).** This dataset [LXL⁺17] contains text passages with an associated multiple-choice question with four possible answer options. The questions were collected from English examinations in China. We use the high school level test set containing 3,498 passage-question sets.

**Winogrande.** This dataset [SBBC19] follows the Winograd schema [LDM11], where the task is a fill-in-a-blank for a given sentence and two options, with additional commonsense reasoning required. We use its test set containing 1,767 questions.

**Wikitext bits/byte.** We evaluate the bits/byte metric using Wikitext2 [MXBS16]. A lower value for this metric implies that less information (bits) is required to make a correct next-token (byte) prediction.

# G   Additional Pseudocode

Previously, in our MIP (Section 4.1), a 1-based indexing system was used. However, for implementation convenience, we use a 0-based indexing system for our pseudocodes instead. Given an ordered sequence $\mathcal{S}$, such as array $A$, string $W$, and selected tokens $\mathbf{S}$, we use $\mathcal{S}_i$ to specify an element in the $i^{th}$ index of $\mathcal{S}$. However, for sequences, we use $\mathcal{S}_{i,j}$ to specify the elements from the $i^{th}$ index up to, but excluding, the $j^{th}$ of $\mathcal{S}$. For example, when $\mathcal{S}$ = happy, we have $\mathcal{S}_1$ = a and $\mathcal{S}_{1,3}$ = ap.

## G.1   Computing $\mathcal{S}$ from $\mathcal{T}$

Given the COUNT function, corpus $\mathbf{W}$, candidate tokens $\mathbf{T}$, and an integer $k$, Algorithm 1 finds a set of tokens $\mathbf{S}$ that maximizes the objective function with the help of subroutines Algorithm 2 and Algorithm 3. The algorithm Algorithm 1 defines a couple of dictionaries $\mathbf{M}$, $\mathbf{P}$, $\mathbf{R}$, and $\mathbf{I}$ to track the problem state, then greedily picks the next best scoring token to cover words:

1. $\mathbf{M}$ maps each word $W$ to its state of cover, similar to the definition in Section 4.1

2. $\mathbf{P}$ maps each token $T$ to the set of its occurrences in the given word $W$, for all $W \in \mathbf{W}$, in a $(W, i)$ pair, where $i$ is the position index of the start of the token occurrence

3. $\mathbf{R}$ stores the net objective gain of each $T$, which we use to greedily select the next best token in Line 8

4. $\mathbf{I}$ maps each token $T$ to an index, which we use to update the state of cover for all $W \in \mathbf{W}$ at Line 18

---

**Algorithm 1** GREEDTOK: Computing $\mathbf{S}$

---

**Require:** COUNT function, corpus words $\mathbf{W}$ where $|W| \geq 2$ for all $W \in \mathbf{W}$, candidate tokens $\mathbf{T}$, integer $k$

1: Initialize dictionary $\mathbf{M} : \mathbf{W} \to \mathbb{N}^+$ with                   ▷ State of the algorithm

$$\mathbf{M}(W) = (m^W_{0,1}, \ldots, m^W_{|W|-2,|W|-1}) = (0, \ldots, 0) = 0^{|W|-1} \qquad \text{for all } W \in \mathbf{W}$$

2: Initialize dictionary $\mathbf{P} : \mathbf{T} \to (\mathbf{W} \times \mathbb{N})^*$ with $\mathbf{P}(T) = \{(W, i) \in \mathbf{W} \times \mathbb{N} : W_{i,i+|T|} = T\}$ for all $T \in \mathbf{T}$

                                          ▷ Positioning information of tokens in words

3: Initialize dictionary $\mathbf{R} : \mathbf{T} \to \mathbb{N}$ with $\mathbf{R}(T) = 0$ for all $T \in \mathbf{T}$      ▷ Token scores given current state

4: Initialize dictionary $\mathbf{I} : \mathbf{T} \to \mathbb{N}$ where $\mathbf{I}(T) = 0$ for all $T \in \mathbf{T}$          ▷ Token indices in $\mathbf{S}$

5: Initialize $\mathbf{S}$ as an empty sequence

6: Compute scores $\mathbf{R}(T)$ for each $T \in \mathbf{T} \setminus \mathbf{S}$ using SCORE on current state $\mathbf{M}$     ▷ Algorithm 3

7: Initialize $\mathbf{Q}$ as a priority queue of $\mathbf{R}$

8: Pop next best token $\tau = \text{argmax}_{T \in \mathbf{T}}$ from $\mathbf{Q}$ and add to back of $\mathbf{S}$ ▷ Skip checking first token

9: **while** $|\mathbf{S}| < k$ **do**

10:      Pop next best token candidate $\hat{\tau} = \text{argmax}_{T \in \mathbf{T}}$ from $\mathbf{Q}$

11:      **if** new score $\hat{\mathbf{R}}(\hat{\tau}) \neq \mathbf{R}(\hat{\tau})$ **then**                             ▷ Algorithm 3

12:          $\mathbf{R}(\hat{\tau}) \leftarrow \hat{\mathbf{R}}(\hat{\tau})$

13:          Push $\hat{\tau}, \mathbf{R}(\hat{\tau})$ back into $\mathbf{Q}$

14:      **else**

15:          Append $\hat{\tau}$ to the back of $\mathbf{S}$ and then update $\mathbf{I}(\hat{\tau}) = |\mathbf{S}|$

16:          **for** $(W, i) \in \mathbf{P}(\tau)$ **do**

17:              **if** CANCOVER$(\mathbf{M}, \tau, W, i)$ **then**

18:                 Update each entry of $\mathbf{M}(W)_{i,i+|\tau|-1}$ to $\mathbf{I}(\tau)$         ▷ Update states of $\mathbf{M}(\tau)$

19:              **end if**

20:          **end for**

21:      **end if**

22: **end while**

23: return $\mathbf{S}$

---

**Algorithm 2** CANCOVER: Check if $W_{i,i+|T|-1}$ is coverable by $T$ in current state $\mathbf{M}$

**Require:** Current state $\mathbf{M}$, token $T$, word $W$, position index $i$
1: **return** ($i = 0$ or $\mathbf{M}(W)_{i-1} = 0$) and ($i + |T| = |W|$ or $\mathbf{M}(W)_{i+|T|-1} = 0$)

---

**Algorithm 3** SCORE: Calculate total number of possible covers

**Require:** Token $T$, token positions $\mathbf{P}(T)$, current state $\mathbf{M}$, COUNT function
1: Make a copy $\mathbf{M}'$ of the state $\mathbf{M}$      ▷ The original state remains unchanged
2: Initialize token score $s = 0$
3: **for** $(W, i) \in \mathbf{P}(T)$ **do**
4:      **if** CANCOVER($\mathbf{M}', T, W, i$) **then**
5:          Add COUNT($W$) $\cdot$ |$\{j \in \{i, \ldots, i + |T|\} : \mathbf{M}'(W)_j = 0\}$| to $s$    ▷ Only add score for non-zero entries
6:          Update each entry of $\mathbf{M}'(W)_{i,i+|T|-1}$ to 1      ▷ Mark to avoid double counting; see Example 5
7:      **end if**
8: **end for**
9: **return** $s$

---

The subroutine Algorithm 2 encapsulates a check of the validity of using a given token $T$ to cover $W$ at position $i$, primarily by observing if the non-start/end endpoint positions $i$ and $i + |T|$ were previously covered by some other token previously; if such a token is present, then $T$ cannot cover $W$ at position $i$. Meanwhile, the subroutine Algorithm 3 calculates the score contribution by token $T$, given the current state $\mathbf{M}$, while accounting for previous covers applied from chosen tokens in $\mathbf{S}$.

**Example 4** (Valid coverings and two sample traces). Consider the example where $\mathbf{T} = \{T_1 = \text{pa}, T_2 = \text{ya}, T_3 = \text{ap}\}$ and $\mathbf{W} = \{W_1 = \text{papaya}, W_2 = \text{impact}\}$. Then, we have $\mathbf{P}(T_1) = \{(W_1, 0), (W_1, 2), (W_2, 2)\}$, indicating that the token $T_1$ appears in $W_1$ at positions 0 and 2, and in $W_2$ at position 2. Using SCORE (Algorithm 3) to update $\mathbf{R}$ would yield $\mathbf{R}(\text{pa}) = 3$, $\mathbf{R}(\text{ya}) = 1$, and $\mathbf{R}(\text{ap}) = 1$, so the greedy step Line 8 of Algorithm 1 would first select token $T_1$ to be included into $\mathbf{S}$. Initially, we have $\mathbf{M}(W_1 = \text{papaya}) = (0, 0, 0, 0, 0)$. After selecting $T_1$ into $\mathbf{S}$, we have $\mathbf{M}(W_1) = (1, 0, 1, 0, 0)$. Recalculating the scores using SCORE on the updated state $\mathbf{M}$ would yield $\mathbf{R}(\text{pa}) = 0$, $\mathbf{R}(\text{ya}) = 1$, and $\mathbf{R}(\text{ap}) = 0$, so the token $T_2$ would be selected next. After selecting $T_2$ into $\mathbf{S}$, we have $\mathbf{M}(W_1) = (1, 0, 1, 0, 2)$ because $\mathbf{I}(T_1 = \text{pa}) = 1$ and $\mathbf{I}(T_2 = \text{ya}) = 2$. One can see that the zero and non-zero locations in $\mathbf{M}$ indicate partition and coverage respectively. Now, ignoring the scoring function, let us instead suppose that we selected $T_3 = \text{ap}$, $T_1 = \text{pa}$, and finally $T_2 = \text{ya}$. When we first selected $T_3 = \text{ap}$, the state of $W_1 = \text{papaya}$ will become $M(W_1) = (0, 1, 0, 0, 0)$ with $\mathbf{I}(T_3 = \text{ap}) = 1$. Next, consider the token $T_1 = \text{pa}$ that appears at positions 0 and 2 of the word $W_1$. At position 0, we see that $\mathbf{M}(W_1)_{i+|T|-1=0+2-1=1} = 1 \neq 0$. Meanwhile, at position 2, we have $\mathbf{M}(W_1)_{i-1=2-1} = 1 \neq 0$. Since there is at least one non-start/end endpoint positions already covered by a token, we *cannot* further use $T_1$ in $W_1$. Finally, let us consider using token $T_2 = \text{ya}$, which appears at position 4 of $W_1$. We see that $i > 0$, $\mathbf{M}(W_1)_{i-1} = 0$, and $i + |T_2| < |W_1|$, we can cover $W_1$ with $T_2$ at position 4, resulting in $M(W_1) = (0, 1, 0, 0, 2)$ with $\mathbf{I}(\text{ya}) = 2$. Note that we do not need to check $\mathbf{M}(W_1)_{i+|T_2|-1}$ because $i + |T_2| < |W_1|$.

**Example 5** (State copying and overcounting). Here, we explain why we require a copy of the state in Algorithm 3 to avoid the overcounting of overlapping repeating substrings. Consider the example of $T_1 = \text{aya}$, $W_1 = \text{ayaya}$, and COUNT($W_1$) = 1, where $\mathbf{P}(T_1) = \{(W_1, 0), (W_1, 2)\}$ and $\mathbf{M}(W_1) = \mathbf{M}'(W_1) = (0, 0, 0, 0)$ initially. In this case, we see that $T_1$ would obtain a score of 1 either by covering $W_1$ at position 0 (i.e. **aya**ya) or position 2 (i.e. ay**aya**), but not both positions simultaneously (i.e. ay**aya**). To see how Algorithm 3 ensures this, let us suppose we considered $(W_1, 0)$ then $(W_1, 2)$ in the for loop iteration. As the endpoints of $(W_1, 0)$ are coverable, we update $\mathbf{M}'(W_1)$ to $(1, 1, 0, 0)$. Note that $\mathbf{M}(W_1)$ still remains unchanged as we have yet to confirm that $T_1$ is the next best token $\tau$. With the updated state $\mathbf{M}'$, we see that the next pair $(W_1, 2) \in \mathbf{P}(T_1)$ is an invalid cover since $\mathbf{M}'(W_1)_{2-1=1} = 1 \neq 0$, which prevents an overcounting. We remark that the choice of updating entries to 1 is arbitrary (i.e. any non-zero value will work) and that one can actually avoid explicitly making a copy of the state in implementation by performing checks in an appropriate manner.

**Runtime complexity for computing $\mathcal{S}$.** Each call to CANCOVER (Algorithm 2) runs in $O(1)$ time. Fix an arbitrary iteration of the while loop in Algorithm 1. Each call to SCORE (Algorithm 3) with token $T$ runs in $O(\sum_{W \in \mathbf{W}} |W|)$ time because it iterates through each position in $\mathbf{P}(T)$ once and considers if $T$ is a valid cover for that position. While we update $\mathbf{M}(W)$ during the iteration, due to CANCOVER (Algorithm 2), each index is updated at most once to a non-zero value, i.e. Example 5, resulting in at most $O(\sum_{W \in \mathbf{W}} |W|)$ total number of updates. Therefore, applied across all tokens $T \in \mathbf{T}$, $k$ number of times, Algorithm 1 takes $O(|\mathbf{T}| \cdot k \cdot \sum_{W \in \mathbf{W}} |W|)$ time to compute $\mathcal{S}$. Empirically, we observe that our lazy strategy scales like $\Theta(|\mathbf{T}| \cdot \sum_{W \in \mathbf{W}} |W|)$. Scores can be greedily updated in small batches of next-best candidate tokens (Line 11 in Algorithm 1 is equivalent to batch size 1), which typically suffices to identify the next-best token to add; we do not need to perform $|\mathbf{T}|$ score updates at every iteration. As a result, the cumulative cost over iterations likely remains much smaller than the naive bound, $\sum_{i \in [1,k]} \#updates_i \in O(|\mathbf{T}|)$.

**Additional implementation remarks.** In practice, it is possible to adopt alternative data representations. For example, instead of a dictionary, one could represent $\mathbf{M}$ as a single contiguous array and define a given word $W$ as a position in the array. One could also use a representation of length $|W|$ for each word instead of the $(|W| - 1)$-sized representation discussed in Line 1 and Section 4.1. For example, covering the word $W = \text{papaya}$ by token $T_1 = \text{pa}$ could be represented by $(1, 1, 1, 1, 0, 0)$ instead of $(1, 0, 1, 0, 0)$. However, in the $(1, 1, 1, 1, 0, 0)$ representation, it is impossible to discern a partition and one has to keep track of additional information regarding duplicates of tokens within the same word. Furthermore, one can avoid redundant calculations of $\mathbf{P}$ by tracking and only recalculating the affected $T$ in words covered by the current $\tau$.

## G.2 Tokenizing a text $W$ using $\mathcal{S}$

In Algorithm 4, we describe how to encode a given text $W$ into its token representation using the token set $\mathbf{S}$ from Algorithm 1. First, in Line 1, we initialize a dictionary to map our tokens in $\mathbf{S}$ according to their order of inclusion to $\mathbf{S}$, and then place singleton tokens $\mathbf{B}$ at the back of the sequence. Next, in Line 2, we find all possible token covers of $W$ using tokens in $\mathbf{S}$ and sort them in Line 3 according to their priority $\mathbf{I}$ and a left-to-right ordering in $W$. Using $\mathbf{M}$ to denote which token covers which position index of $W$, we iterate through $(T, i)$ in the sorted $\mathbf{P}$ and update $\mathbf{M}$ whenever the token $T$ can cover $W$ at position $i$ given earlier decisions. Note that this may mean that a later token of longer length may overwrite the covering decision of an earlier shorter token; see Example 6. Finally, using $\mathbf{M}$, we return the 0-delineated token representation; see Example 7.

---

**Algorithm 4** GREEDTOK: Tokenizing a given text $W$ using $\mathbf{S}$

---

**Require:** Text $W$, singleton tokens $\mathbf{B}$, chosen token sequence $\mathbf{S}$
1: Initialize dictionary $\mathbf{I} : \mathbf{S} \cup \mathbf{B} \to \mathbb{N}$ with

$$\mathbf{I}(T) = \begin{cases} i & \text{if } T \text{ is } i^{th} \text{ chosen token in } \mathbf{S} \\ |\mathbf{S}| + i & \text{if } T \text{ is } i^{th} \text{ token in } \mathbf{B} \end{cases}$$

$\qquad\qquad\qquad\qquad\qquad\qquad\qquad\qquad$ ▷ Fix an arbitrary ordering to singleton tokens
2: Initialize potential cover positions $\mathbf{P} = \{(T, i) : T \in \mathbf{S}, W_{i, i+|T|} = T\}$
3: Sort $\mathbf{P}$ based on $\mathbf{I}(T)$, then position $i$, with lower values having greater priority
4: Initialize state $\mathbf{M} = \{m_{0,1}, \ldots, m_{|W|-2, |W|-1}\} = 0^{|W|-1}$
5: **for** $(T, i) \in \mathbf{P}$ in descending sorted order of Line 3 **do**
6: $\quad$ **if** CANCOVER$(\mathbf{M}, T, W, i)$ **then**
7: $\quad\quad$ Update each entry of $\mathbf{M}_{i, i+|T|-1}$ to $\mathbf{I}(T)$
8: $\quad$ **end if**
9: **end for**
10: **return** $W$ delineated at positions of $0$ $\qquad\qquad\qquad\qquad\qquad$ ▷ See Example 7

---

**Example 6** (Overriding earlier shorter tokens)**.** Consider the encoding of $W_1 = \text{abcdefg}$ with $\mathbf{S} = (S_1 = \text{ab}, S_2 = \text{cd}, S_3 = \text{ef}, S_4 = \text{abc}, S_5 = \text{abcd}, S_6 = \text{efg}, S_7 = \text{abcdefg})$. In the first three iterations, we use $S_1$, $S_2$, and $S_3$ to cover $W$, resulting in $\mathbf{M} = (1, 0, 2, 0, 3, 0)$. Then, we see that $S_4$ does not have any valid covers and so $\mathbf{M}$ remains unchanged. In the fifth and sixth iterations, notice that $S_1, S_2 \subset S_5$ and $S_3 \subset S_6$, resulting in $S_5$ and $S_6$ being valid covers and $\mathbf{M}$ being updated

to $(5, 5, 5, 0, 6, 6)$. Finally, since $S_5, S_6 \subset S_7$ and $S_7$ is a valid cover with respect to the current state, $\mathbf{M}$ becomes $(7, 7, 7, 7, 7, 7)$. Now, consider another scenario of encoding $W_2 = \text{abcd}$ using $\mathbf{S} = (S_8 = \text{ab}, S_9 = \text{abc}, S_{10} = \text{abcd})$, where $S_8 \subset S_9 \subset S_{10}$. Covering $W_2$ using $S_8$ results in $\mathbf{M} = (1, 0, 0)$. Then, using $S_9$ results in $\mathbf{M} = (2, 2, 0)$. Finally, using $S_{10}$ results in $\mathbf{M} = (3, 3, 3)$. In both examples, we see that covers are only overridden by proper supersets that appear later in the ordering of $\mathbf{S}$, where the largest valid cover in $\mathbf{S}$ for $W$ is of size $|W|$. Furthermore, recall that the token covers of any valid covering do not overlap so they jointly take up at most $|W|$ positions in total. As such, we see that each position $\mathbf{M}_i \in \mathbf{M}$ is updated at most $|W|$ times and thus, across all $|W|$ positions, Algorithm 3 updates values in $\mathbf{M}$ a maximum of $O(|W|^2)$ times.

**Example 7** (Encoding the tokenized output). If $W = \text{abcdef}$, $\mathbf{S} = \{S_1 = \text{bcd}, S_2 = \text{ef}\}$ and $\mathbf{M} = (0, 1, 1, 0, 2)$, then $W$'s final tokenized output will be $(\text{a}, \text{bcd}, \text{ef})$. If one wishes to convert the tokens to integers with respect to token indexing, simply apply $\mathbf{I}$ to each token to get $(\mathbf{I}(\text{a}), \mathbf{I}(\text{bcd}), \mathbf{I}(\text{ef}))$.

**Runtime complexity for tokenizing $W$ using $\mathcal{S}$.** Each call to CANCOVER (Algorithm 2) runs in $O(1)$ time. There are at most $\binom{|W|}{2} \in O(|W|^2)$ substrings of $W$ and so Line 2 runs in $O(|W|^2)$ time, $|\mathbf{P}| \in O(|W|^2)$ and sorting $\mathbf{P}$ takes $O(|W|^2 \log(|W|))$ time. Since each index can only be overwritten when a longer token covers it, in Line 7, we see that each position in $\mathbf{M}$ is only updated at most $|W|$ times, and therefore a maximum of $|W|^2$ for all positions in $|\mathbf{P}|$ number of iterations; see Example 6. Thus, the entire for loop takes $O(|\mathbf{P}| + |W|^2) \subseteq O(|W|^2)$ time to iterate through $\mathbf{P}$ and to update all $O(|W|)$ positions in $\mathbf{M}$.

**Additional implementation remarks.** In practice, we limit the subsequence search to the maximum token length $\ell = \max_{T \in \mathbf{S}} |T|$, with early stopping. To reduce $|W|$ even further, we have to go beyond regex and identify smaller local sections within $W$ so that we can independently tokenize these sections. This is possible as $\mathbf{S}$ inadvertently learns the regex pattern and more during its construction. This implies that we can also further infer natural separations within $W$ where no $T \in \mathbf{S}$ overlaps with another.

