# OpenReview forum: "A Partition Cover Approach to Tokenization"
_NeurIPS.cc/2025/Conference — NeurIPS 2025 poster_

### Official Review · Reviewer_MQ49 · 2025-06-22

**Clarity:** 3
**Significance:** 3
**Originality:** 3
**Rating:** 4
**Confidence:** 4

**Summary:**

This paper studies the computational complexity of the tokenization problem and proves that it is NP-hard. The authors establish this result by reducing the original token search problem to a tokenization decision problem, which they show to be NP-hard via a reduction from the Vertex Cover problem. Constrainet optimization-based formulation is also introduced.

To address the intractability, the paper proposes a new approximation algorithm designed to improve compression quality while enjoying a polynomial time complexity. Experimental results on real-world datasets demonstrate that the proposed method outperforms existing approaches such as BPE and Unigram in terms of compression/perplexity performance.

**Questions:**

It would be helpful to clarify whether there is any meaningful distinction between the problem formulation presented in Problem 1 and Definition 2 in [WBP24], especially since this work aims to establish analogous results. A direct comparison or discussion would strengthen the exposition and help the reader appreciate the novelty or alignment of the proposed formulation with prior work.

Line 261 notes that preprocessing is required to extract words from the corpus. Could the authors comment on whether their algorithm can be adapted to work without such preprocessing? This would align more closely with the setting considered by SuperBPE [1], which, to my understanding, also reflects the original BPE algorithm without additional preprocessing steps. Addressing this would help clarify the scope and flexibility of the proposed approach.

Line 237-243, is it fair to say that GreedTok is an extension of BPE but without pairwise merge constraints, i.e., we are allowed to merge any $n$ tokens to create a new one? How optimal is the proposed algorithm compares to other recent tokenization scheme such as [SRZ+24]?

Is there any insights why there is little difference in terms of performance in Figure 2b? For tokens/words $t$ that occurs in both vocabularies, is there any significant different in terms of the empirical next-token probability $P(t | prompt')$?

[1] Liu, Alisa, et al. "Superbpe: Space travel for language models." arXiv preprint arXiv:2503.13423 (2025).

**Ethical Concerns:**

["NO or VERY MINOR ethics concerns only"]

**Final Justification:**

This paper studies the cover problem of tokenization and provide some interesting theoretical contribution.

**Limitations:**

The paper currently lacks clarity in several foundational definitions, making it difficult to follow. For example, in Section 2, the content of the alphabet $\Sigma$ is not explicitly defined. Additionally, the notation $\Sigma^+$ is used without explanation—what does it represent, and how does it relate to $\Sigma$?

It is also unclear why $\Sigma^+$ is assumed to contain \emph{words}, and this assumption is neither justified nor motivated. Further confusion arises from inconsistent usage: $\Sigma^+$ is described as containing \emph{words} on line 89 and \emph{tokens} on line 97. Are \emph{tokens} and \emph{words} considered equivalent in this context, or is there a meaningful distinction between them? Clarifying these points would significantly improve the precision and readability of the paper. In fact, I have to jump back and forth between Section 2,3,4 to better understand the notations.

Empirical/Theoretical runtime complexity should be included in the paper as a table of reference for practitioner.

**Quality:**

3

**Strengths And Weaknesses:**

Strengths:
- The paper provides an alternative algorithmic viewpoint on the complexity of tokenization.
- The proposed tokenization scheme outperforms the standard baseline

Weakness:
- [Major] The writing is confusing (see limitations).
- [Minor] No approximation guarantee for GreedTok.

---

> ### Author Rebuttal · Authors · 2025-07-30
>
> Thank you for taking the time to review our work and for the thoughtful and constructive feedback. We address your questions and comments below.
>
> ### Comparison with [WBP24]
>
> We appreciate the suggestion to clarify our relation to [WBP24]. Problem 1 in our paper is indeed equivalent to Definition 2 in [WBP24], up to notational differences. We will make this equivalence explicit in the revised version.
>
> However, our work differs from [WBP24] in two key aspects. Firstly, [WBP24] proves NP-hardness via a lengthy reduction from MAX-2-SAT while we offer a more concise 1-page reduction from vertex cover. Moreover, [WBP24] does not propose an algorithm or report empirical results, whereas our work introduces GreedTok and validates it through experiments on real-world NLP datasets. We believe this adds both theoretical and practical contributions beyond prior work.
>
> ### Pre-processing and relation to BPE
>
> GreedTok is not an extension of BPE. It is a fundamentally different construction that does not rely on pairwise merges. While it is true that GreedTok allows more general n-wise merges, it does not inherit the inductive structure of BPE and therefore cannot operate without preprocessing. This behavior is more similar to Unigram tokenization, which also requires word-level frequency statistics prior to training. That said, GreedTok's preprocessing step can be accelerated by leveraging prior knowledge of high-quality token candidates (e.g., frequent substrings or n-grams), or by restricting candidate generation based on token length or frequency. We will clarify this in the paper.
>
> ### Comparison with SuperBPE [1]
>
> SuperBPE [1] merges tokens across whitespace boundaries to form "super tokens" after an initial BPE pass. This approach leads to larger token sets and can complement GreedTok's output. For instance, GreedTok tokens can be used as the base vocabulary from which SuperBPE-style merges are subsequently applied. We consider this a promising direction for future work. Since SuperBPE appeared in March 2025 (contemporaneous with our submission), we did not include it in the initial comparison but will mention it in our revision.
>
> ### Comparison against PathPiece [SRZ+24]
>
> PathPiece [SRZ+24] is a decoding algorithm that operates independently of how the vocabulary is constructed. It assumes a pre-defined token set and does not perform token selection or training. As such, PathPiece can be directly applied to any token vocabulary, including one generated by GreedTok. Our focus is on the vocabulary construction step, and we see PathPiece as a complementary technique rather than a competing one.
>
> ### Performance insights and Figure 2b
>
> Thank you for the insightful question regarding the results in Figure 2b. GreedTok produces a more compact representation of text (i.e., fewer tokens per sentence), which influences the average log-likelihood per token. For example, while the BPE-trained model may achieve an average negative log-likelihood of ~2.0 per token, GreedTok-trained models may show ~2.5, due to fewer, longer tokens. However, directly evaluating on this metric alone can be misleading. To fairly compare models with different token granularities, we evaluate performance in terms of bits-per-byte. This metric is independent of tokenization and reflects true compression performance on the underlying data. Figure 2b shows that, when normalized in this way and trained on equivalent byte-lengths of data, GreedTok performs comparably to BPE. This highlights the competitive modeling capacity of GreedTok despite structural differences in tokenization. We will add a version of this discussion in our revision.
>
> ### Notation concerns
>
> Below, we clarify our notation and we will add a version of this discussion to our revision.
>
> * Alphabet ($\mathbf{\Sigma}$) refers to the basic character set. This may be the 26 lowercase English letters or the full Unicode set, depending on context.
>
> * The star superscript is standard Kleene-star notation in Computer Science to mean "0 or more times", which is also commonly used in Regex expressions. In our context, $\mathbf{\Sigma}^*$ denotes the set of all finite strings formed from characters in $\mathbf{\Sigma}$.
>
> * $\mathbf{W}$ refers to the set of distinct words in the corpus, which is why we have $\mathbf{W} \subseteq \mathbf{\Sigma}$. Menawhile, tokens $\mathbf{T}$ are candidate substrings drawn from $\mathbf{\Sigma}^*$, that may or may not correspond to actual words.
>
> ### Runtime complexity
>
> Our theoretical runtime analysis is presented in Section 6 (Page 9, Lines 347–358) and Appendix G. Meanwhile, our empirical timings are reported in Table 1 (Page 6).
>
> ### References
>
> [1] Alisa Liu and Jonathan Hayase and Valentin Hofmann and Sewoong Oh and Noah A. Smith and Yejin Choi. SuperBPE: Space Travel for Language Models. arXiv, 2025.

---

> > ### Comment · Reviewer_MQ49 · 2025-08-03
> >
> > Thank you for your feedback, the paper is more clear to me now. I will maintain my score.

---

### Official Review · Reviewer_gs2o · 2025-07-02

**Clarity:** 4
**Significance:** 3
**Originality:** 3
**Rating:** 5
**Confidence:** 4

**Summary:**

This paper proposes a novel method for tokenization. The authors first show that the tokenization problem can be reduced from the vertex cover problem, showing that it is an NP-hard optimization problem. They then propose a greedy optimization algorithm to address it. Since the relaxed version of the problem corresponds to the weighted maximum coverage problem, greedy algorithms are expected to yield good solutions. Through experiments on multiple datasets, the authors demonstrate that their method achieves higher compression rates than BPE and Unigram. Furthermore, when used for language model pre-training, the proposed method outperforms BPE on various downstream benchmarks.

**Questions:**

- While lazy evaluation strategies are well known to be fast in practice, I think their exact cost is unknown. How do you drive  $\Theta(|T|\sum_{W}|W|)$ in line 350?  If you aim to speed up the greedy algorithm, applying the stochastic greedy algorithm [2] may offer further speedups.

- Why is the proposed method not compared against UNIGRAM in the experiments presented in Section 5.2?


[2] Baharan Mirzasoleiman, Ashwinkumar Badanidiyuru, Amin Karbasi, Jan Vondrák, and Andreas Krause. 2015. Lazier than lazy greedy. In Proceedings of the Twenty-Ninth AAAI Conference on Artificial Intelligence (AAAI'15). AAAI Press, 1812–1818.

**Ethical Concerns:**

["NO or VERY MINOR ethics concerns only"]

**Final Justification:**

The authors have adequately addressed my concerns during the discussion, so I will keep my initial score. I believe this paper presents a theoretically grounded, simple, and novel method that is also effective in practice.

**Limitations:**

Yes

**Quality:**

4

**Strengths And Weaknesses:**

# Strengths
- This paper reveals a connection between tokenization and the vertex cover problem and presents a novel tokenization method from a fresh perspective, distinct from existing approaches. The proposed method is original and also appealing in its simplicity.

- The paper is clearly written and includes many illustrative examples, making it easy to follow.

- Experimental results show the advantage of the proposed method over standard tokenization baselines.

- While no theoretical guarantee is provided for the solution quality, the weighted maximum coverage problem is well known to admit good approximations with greedy algorithms. The proposed method empirically achieves strong performance in this regard.

# Weaknesses
- As a method for selecting an optimal token dictionary, VOLT [1] is known to be effective. VOLT has been reported to outperform SentencePiece and BPE on machine translation tasks. Although I appreciate the simplicity of the proposed method, it would be preferable to include a comparison with VOLT or a discussion of the advantages of the proposed method relative to it.

[1] Xu et al. Vocabulary Learning via Optimal Transport for Neural Machine Translation, ACL-IJCNLP 2021. https://aclanthology.org/2021.acl-long.571.pdf

---

> ### Author Rebuttal · Authors · 2025-07-30
>
> Thank you for taking the time to review our work and for the thoughtful and constructive feedback. We address your questions and comments below.
>
> ### Comparison with VOLT [1]
>
> We appreciate your suggestion to consider VOLT [1] in our evaluation. VOLT is an effective method for selecting a token vocabulary by maximizing the entropy of subword distributions across a candidate set. It operates by selecting from an existing pool of tokens generated via Unigram or BPE, often with variable vocabulary sizes. While VOLT excels at vocabulary selection, it is complementary to token generation methods such as GreedTok.
>
> In fact, GreedTok could serve as an upstream tokenizer to generate candidate tokens for VOLT, especially when the target vocabulary size is flexible and full character coverage is required. Our focus in this work was on building a standalone token generation method with strong compression and practical usability under fixed-size constraints. Nonetheless, we agree that exploring a GreedTok-then-VOLT pipeline could be an interesting direction for future work, and we will include this discussion in the revised paper.
>
> ### Lazy evaluation and computational cost
>
> Thank you for highlighting the need for a clearer explanation of the computational cost of our lazy evaluation strategy. Our bound of $|\mathbf{T}| \cdot \sum_{W} |W|$ is not an actual analytical bound but an empirically observed one. We will make correct the phrasing and make it clear in our revision.
>
> In practical scenarios, the target vocabulary size $k$ is much smaller than the number of candidate tokens $|\mathbf{T}|$, which can be upwards of 100M. Despite this, we observe empirically that it is rarely necessary to evaluate all $|\mathbf{T}|$ token scores in each iteration. In our implementation, we update scores in small batches of 100 candidate tokens, which typically suffices to identify the next best token to add. In some iterations, score updates can be entirely skipped due to caching from previous steps. As a result, the cumulative cost over $k$ iterations remains much smaller than the naive $k \cdot |\mathbf{T}|$ bound. That is, we observe that our lazy strategy scales *empirically* like $|\mathbf{T}| \cdot \sum_{W} |W|$.
>
> ### Stochastic greedy algorithm of [2]
>
> We appreciate the suggestion to consider stochastic greedy algorithms, such as [2], as a potential speedup technique. This is indeed an intriguing direction since stochastic methods could further reduce evaluation overhead by sampling candidates rather than scanning deterministically. While not explored in our current work, this aligns well with the goals of GreedTok, and we are excited by the prospect of exploring such techniques. We will add this as a promising avenue for future research.
>
> ### On the absence of Unigram in Section 5.2
>
> We appreciate your interest in seeing Unigram included in our evaluation. There were several reasons why we ultimately chose not to include it in Section 5.2.
>
> As noted in prior work [3, 4], HuggingFace's (HF) implementation of Unigram has some known limitations. For example, it deviates from the original SentencePiece implementation and exhibits certain biases (see Footnote 17 in [3]). More critically, the HF Unigram tokenizer lacks support for byte-level fallback, resulting in rare or unknown inputs being mapped to `<unk>` tokens, rather than a more robust byte-level representation. This makes a fair comparison with BPE difficult, as BPE does offer byte-level fallback. We attempted to simulate byte-level fallback in HF Unigram by appending ASCII-escaped byte tokens to the vocabulary. However, this resulted in many tokens being encoded as escaped Unicode strings (e.g., `\\x...`), which degraded performance.
>
> As for the original SentencePiece Unigram implementation, we encountered further difficulties. Training it on the full 20\% DCLM corpus used for our tokenizer training exceeded our available memory capacity. Reducing the training data to fit memory constraints would have significantly weakened compression quality, further limiting the validity of comparisons. Additionally, SentencePiece Unigram does not allow for precise control over vocabulary size as it tends to add extra characters as tokens after training. This resulted in differing token counts and thus models with different parameter sizes.
>
> Given these challenges and our finite compute resources, we chose to focus on the BPE versus GreedTok comparison, which allows for cleaner and more reproducible evaluation. Moreover, BPE remains the most widely adopted method in current practice, making the comparison more directly relevant. That said, we agree that a comparison with Unigram would be interesting. If you are aware of better-supported or alternative implementations of Unigram with byte-level fallback, we would be very interested in exploring them. Thank you again for this suggestion.
>
> ### References
>
> [1] Jingjing Xu, Hao Zhou, Chun Gan, Zaixiang Zheng, and Lei Li. Vocabulary Learning via Optimal Transport for Neural Machine Translation. ACL, 2021.
>
> [2] Baharan Mirzasoleiman, Ashwinkumar Badanidiyuru, Amin Karbasi, Jan Vondrák, and Andreas Krause. 2015. Lazier than lazy greedy. In Proceedings of the Twenty-Ninth AAAI Conference on Artificial Intelligence (AAAI'15). AAAI Press, 1812–1818.
>
> [3] Craig W Schmidt, Varshini Reddy, Haoran Zhang, Alec Alameddine, Omri Uzan, Yuval Pinter, and Chris Tanner. Tokenization Is More Than Compression. ACL, 2024.
>
> [4] Mehdi Ali, Michael Fromm, Klaudia Thellmann, Richard Rutmann, Max Lübbering, Johannes Leveling, Katrin Klug, Jan Ebert, Niclas Doll, Jasper Buschhoff, Charvi Jain, Alexander Weber, Lena Jurkschat, Hammam Abdelwahab, Chelsea John, Pedro Ortiz Suarez, Malte Ostendorff, Samuel Weinbach, Rafet Sifa, Stefan Kesselheim, and Nicolas Flores-Herr. Tokenizer Choice For LLM Training: Negligible or Crucial?. NAACL, 2024.

---

> > ### Comment · Reviewer_gs2o · 2025-08-03
> >
> > Thank you for the response. The authors have adequately addressed my concerns. I will maintain my review score.

---

### Official Review · Reviewer_58Hv · 2025-07-04

**Clarity:** 4
**Significance:** 2
**Originality:** 2
**Rating:** 5
**Confidence:** 4

**Summary:**

The paper introduces a new optimization-based formulation for the tokenization problem in NLP, generalizing beyond the standard merge-based algorithms like Byte-Pair Encoding (BPE). The authors formulate tokenization in a more general form as a partition cover optimization problem and argue that optimal tokenization is NP-hard providing a clearer intuituve arguments. Then they propose GREEDTOK, a polynomial-time greedy algorithm for token selection, and empirically show it outperforms BPE and Unigram in terms of compression on real-world corpora.

The authors also try to demonstrate practical downstream impact but that part of the paper is less convincing to me. Based on the reported experiments, GREEDTOK achieves better compression and equal or improved language modeling performance on several benchmarks.

**Questions:**

Could you provide some futher analysis of token quality?

Could you elaborate on why you only compare the proposed method with two relatively old ones (the inefficiencies and flaws of BPE and Unigram are well documented in the literature, after all)?

**Ethical Concerns:**

["NO or VERY MINOR ethics concerns only"]

**Final Justification:**

After reading authors' response I have decided to update my final evaluation to 5.

**Limitations:**

I generally like the paper, but I am very concered about statistical significance of the proposed results (especially for downstream tasks). I think that authors could have addressed the limitation quality themselves in a more comprehensive manner.

**Quality:**

3

**Strengths And Weaknesses:**

The paper has several good aspectes that clearly speak in its favour:

1. The partition cover formulation is an interesting conceptual advance

2 I found the NP-hardness discussion vetu elegant. The connection to weighted maximum coverage is both theoretically and practically valuable.

3 The proposed GREEDTOK consistently achieves better compression than BPE and Unigram

4 The paper provides detailed pseudocode, experimental protocols, and ablation studies, facilitating reproducibility.

Since tokenization is a foundational problem for large language models improvements here have broad implications for efficiency and model capability.

There are also some of the issues I have with the presented paper.

First and formost, error bars or statistical significance are not provided for downstream tasks, though this is understandable given the resource requirements, in my opinion it significantly undermines the practical importance of the proposed algorithm. This is further enhanced due to the absence of some comparison with various optimised versions of BPE, such as BPE-dropout (https://aclanthology.org/2020.acl-main.170/), PickyBPE (https://aclanthology.org/2024.emnlp-main.925/) or Boundless BPE (https://arxiv.org/abs/2504.00178). On top of that, while compression is shown to improve, the analysis of token quality (e.g., semantic coherence, impact on multilingual or morphologically rich languages) is less extensive than some recent studies.

---

> ### Author Rebuttal · Authors · 2025-07-30
>
> Thank you for taking the time to review our work and for the thoughtful and constructive feedback. We address your questions and comments below.
>
> ### Error bars and statistical significance
>
> We agree that reporting error bars or significance testing would further strengthen our results. However, as noted in prior works [1, 2], it is common practice in large-scale language modeling to train only one model per tokenizer variant due to the high computational cost. In our case, we followed a similar approach: rather than training many smaller models for variance estimates, we chose to allocate our compute budget to training a single strong model for each tokenizer using established best practices for BPE-based models [3], each on 500–600 billion tokens. We acknowledge this as a limitation of our study and will make this more explicit in the revised paper.
>
> ### Comparison with other variants of BPE
>
> Thank you for pointing out recent advances in BPE tokenization, such as [4,5,6]. We agree that these are important and relevant works and appreciate the opportunity to clarify their relationship to our method.
>
> BPE-Dropout [4] introduces stochasticity by randomly dropping merge operations during training, yielding multiple possible segmentations per input. While GreedTok is deterministic by default, it can be adapted in a similar fashion: at each step, we can randomly skip adding the top token and proceed with updating the graph accordingly.
>
> PickyBPE [5] refines the BPE vocabulary by iteratively removing low-utility tokens, using a deletion criterion guided by a hyperparameter. Encoding then proceeds using greedy decoding or PathPiece [1]. GreedTok can likewise incorporate such retrospective pruning: after each token addition, evaluate and remove earlier redundant tokens to improve vocabulary efficiency.
>
> Boundless BPE [6] allows token merging across whitespace boundaries by generating super tokens after an initial round of BPE training. While this results in larger token sets, this idea is complementary to GreedTok. In fact, GreedTok tokens could be used as an initial set from which whitespace-spanning merges are constructed. We consider this direction interesting and note that Boundless BPE appeared on arXiv in March 2025, contemporaneous with our own submission.
>
> Overall, we see GreedTok as a flexible base algorithm that could be extended in future work with ideas from these recent works. Hence, we compare GreedTok with other base tokenizer algorithms, BPE/Unigram, focusing on the core tokenizer algorithm implementations without extra features or extensions.
>
> ### Analysis of token quality
>
> We appreciate your request for a deeper analysis of token quality. While our main focus was on compression and downstream performance, we did conduct some additional investigations, which we summarize below and will expand in the revision.
>
> *Morphological coherence*: To assess whether GreedTok captures meaningful subword units, we compared token sets on the coverage of English prefixes and suffixes (using lists from Wikipedia-EN). We found that GreedTok performs comparably to BPE in capturing these meaningful affixes; see table below.
>
> Table: Number and \% of relevant suffixes/prefixes found
> |    |GreedTok |  BPE | Unigram|
> | --- |:---: | :---: | :---: |
> |*Prefix* | 142 (36.1\%) | 146 (37.2\%) | 157 (39.9\%)|
> |*Suffix* | 112 (26.0\%) | 112 (26.0\%) | 94 (21.9\%)|
>
> *Multilingual performance*: We analyzed tokenization behavior on Wikipedia data in Chinese, Japanese, and Korean (CJK), both individually and jointly (experiment mentioned around line 352). We observed that GreedTok tends to introduce longer tokens, often corresponding to full nouns, earlier as compared to BPE which enforces pairwise merges. In CJK languages, phonetic loanwords from English are often translated into sequences of characters that do not carry individual semantic meaning. GreedTok naturally merges such sequences into single tokens, which can be desirable as the individual characters in such sequences do not carry semantic meaning. Below, we list some examples where pairwise merging is bypassed, with \# indicating the order of inclusion in the token set:
>
>
> *Chinese*
>
> 平方公里 - square kilometre (12 bytes) (12 way merge from bytes) \#123
>
> 华人民共和国 - People’s Republic of China (18 bytes, 6 way merge) \#551
>
> 洛杉磯 - Los Ángeles (9 bytes) (7 way merge; first token merged before, rest from bytes) \#4229
>
> 迪士尼 - Disney (9 bytes; 3 way merge) \#6684
>
> *Japanese*
>
> プログラム - program  (15 bytes, 3-way merge) \#2844
>
> ロンドン - London (12 bytes, 3-way merge) \#2857
>
> ロサンゼルス - Los Ángeles (18 bytes) (4-way merge) \#7066
>
> ディズニー - Disney (15 bytes, 3-way merge) \#7980
>
> *Korean*
>
> 로스앤젤레스 - Los Ángeles (18 bytes, 6-way merge) \#1630
>
> 엔터테인먼트 - entertainment (18 bytes, 6-way merge) \#1740
>
> 디즈니 - Disney (9 bytes, 3-way merge) \#6606
>
> ### References
>
> [1] Craig W Schmidt, Varshini Reddy, Haoran Zhang, Alec Alameddine, Omri Uzan, Yuval Pinter, and Chris Tanner. Tokenization Is More Than Compression. ACL, 2024.
>
> [2] Mehdi Ali, Michael Fromm, Klaudia Thellmann, Richard Rutmann, Max Lübbering, Johannes Leveling, Katrin Klug, Jan Ebert, Niclas Doll, Jasper Buschhoff, Charvi Jain, Alexander Weber, Lena Jurkschat, Hammam Abdelwahab, Chelsea John, Pedro Ortiz Suarez, Malte Ostendorff, Samuel Weinbach, Rafet Sifa, Stefan Kesselheim, and Nicolas Flores-Herr. Tokenizer Choice For LLM Training: Negligible or Crucial?. NAACL, 2024,
>
> [3] Mayank Mishra. Dolomite Engine: A Hyper-Optimized Library for Pretraining and Finetuning. Github, 2024.
>
> [4] Ivan Provilkov, Dmitrii Emelianenko, Elena Voita. BPE-Dropout: Simple and Effective Subword Regularization. ACL, 2020.
>
> [5] BPE Gets Picky: Efficient Vocabulary Refinement During Tokenizer Training
> Pavel Chizhov, Catherine Arnett, Elizaveta Korotkova, Ivan Yamshchikov. BPE Gets Picky: Efficient Vocabulary Refinement During Tokenizer Training. ACL, 2024
>
> [6] Craig W. Schmidt, Varshini Reddy, Chris Tanner, Yuval Pinter. Boundless Byte Pair Encoding: Breaking the Pre-tokenization Barrier. arXiv, 2025.

---

### Official Review · Reviewer_9YQ1 · 2025-07-23

**Clarity:** 4
**Significance:** 3
**Originality:** 3
**Rating:** 5
**Confidence:** 2

**Summary:**

The authors introduce GreedTok, a polynomial-time greedy tokenization algorithm which draws on theory from the weighted maximum coverage problem. They show GreedTok outperforms byte pair encoding on compression while more closely resembling Unigram-like tokens, and run pretraining experiments that show models using GreedTok outperforms BPE accuracy when configured to use the same number of tokens as BPE.

**Questions:**

- There is some discussion of GreedTok having a more closely aligned tokenization to Unigram. Unigram is left out of the Table 4 evaluation on accuracy across popular benchmarks, presumably for computational reasons. However, it would be interesting to see how Unigram compares, even if on a smaller scale, to both BPE and GreedTok. This may further validate your hypothesis that performance gains are because GreedTok “may inherit some of UNIGRAM’s favorable token characteristics”
- Any evidence of how this tokenization would do in terms of robustness to adversarial attacks? BPE is a strong defense against text perturbation attacks which attempt to produce out-of-vocabulary tokens. Presumably GreedTok achieves this as well due to its high compression rate, but it is never discussed explicitly.

**Ethical Concerns:**

["NO or VERY MINOR ethics concerns only"]

**Limitations:**

- Limited to 1B parameter models, though due to computational costs this is reasonable
- No exploration of low-resource languages

**Paper Formatting Concerns:**

- None, the paper is well formatted.

**Quality:**

3

**Strengths And Weaknesses:**

Strengths
- Clear writing and well structured paper
- Experimental results comparing performance when using GreedTok vs BPE are compelling and interesting
- The method is practically viable for full-scale modern NLP workloads
- Appendix contains many important details for replicability as well as further interesting findings and analysis

Weaknesses
- While I am not an expert in theory and algorithms, the theoretical contribution appears sound to me and is backed up by empirical experiments
- The algorithm is more computationally complex than BPE, although this is not too bad given it amounts to a one-time cost upfront that does not affect future model training runs

---

> ### Author Rebuttal · Authors · 2025-07-30
>
> Thank you for taking the time to review our work and for the thoughtful and constructive feedback. We address your questions and comments below.
>
> ### On the absence of Unigram in Table 4
>
> We appreciate your interest in seeing Unigram included in our evaluation. There were several reasons why we ultimately chose not to include it in Table 4.
>
> As noted in prior work [1, 2], HuggingFace's (HF) implementation of Unigram has some known limitations. For example, it deviates from the original SentencePiece implementation and exhibits certain biases (see footnote 17 in [1]). More critically, the HF Unigram tokenizer lacks support for byte-level fallback, resulting in rare or unknown inputs being mapped to `<unk>` tokens, rather than a more robust byte-level representation. This makes a fair comparison with BPE difficult, as BPE does offer byte-level fallback. We attempted to simulate byte-level fallback in HF Unigram by appending ASCII-escaped byte tokens to the vocabulary. However, this resulted in many tokens being encoded as escaped Unicode strings (e.g., `\\x...`), which degraded performance.
>
> As for the original SentencePiece Unigram implementation, we encountered further difficulties. Training it on the full 20\% DCLM corpus used for our tokenizer training exceeded our available memory capacity. Reducing the training data to fit memory constraints would have significantly weakened compression quality, further limiting the validity of the comparisons. Additionally, SentencePiece Unigram does not allow for precise control over vocabulary size as it tends to add extra characters as tokens after training. This resulted in differing token counts and thus models with different parameter sizes.
>
> Given these challenges and our finite compute resources, we chose to focus on the BPE versus GreedTok comparison, which allows for cleaner and more reproducible evaluation. Moreover, BPE remains the most widely adopted method in current practice, making the comparison more directly relevant. That said, we agree that a comparison with Unigram would be interesting. If you are aware of better-supported or alternative implementations of Unigram with byte-level fallback, we would be very interested in exploring them. Thank you again for this suggestion.
>
> ### Robustness to adversarial attack
>
> From our understanding, adversarial attacks in the context of tokenization often rely on producing non-canonical token sequences that attempt to evade alignment or safety filters.
> Thus, the effectiveness of such attacks, and the robustness of a model against them, ultimately depend not just on the tokenizer but also on how the downstream model processes the resulting tokens. We believe that GreedTok confers similar robustness properties as BPE as long we consider the same adversarial setup, and adversarial robustness is an interesting angle to explore in future work.
>
> ### Limitations
>
> We appreciate your observations and agree with both points.
>
> While our experiments were limited to 1B-parameter models due to resource constraints, we believe the insights and trends observed should generalize to larger models, though we acknowledge that empirical confirmation at scale is needed. We will clarify this point in the revised paper.
>
> Regarding low-resource languages, we recognize this as an important area for further exploration. As we are not experts in low-resource languages, we refrained from making claims about generalization to other linguistic settings. We will explicitly acknowledge this limitation and note it as a valuable direction for future work, ideally in collaboration with domain experts in multilingual and low-resource NLP.
>
> ### References
>
> [1] Craig W Schmidt, Varshini Reddy, Haoran Zhang, Alec Alameddine, Omri Uzan, Yuval Pinter, and Chris Tanner. Tokenization Is More Than Compression. ACL, 2024.
>
> [2] Mehdi Ali, Michael Fromm, Klaudia Thellmann, Richard Rutmann, Max Lübbering, Johannes Leveling, Katrin Klug, Jan Ebert, Niclas Doll, Jasper Buschhoff, Charvi Jain, Alexander Weber, Lena Jurkschat, Hammam Abdelwahab, Chelsea John, Pedro Ortiz Suarez, Malte Ostendorff, Samuel Weinbach, Rafet Sifa, Stefan Kesselheim, and Nicolas Flores-Herr. Tokenizer Choice For LLM Training: Negligible or Crucial?. NAACL, 2024.

---

> ### Comment · Reviewer_9YQ1 · 2025-08-06
> **Response to authors**
>
> Thank you for the response. I appreciate your detailed reply regarding the challenges of a unigram model, and comments on low-resource evaluation. I have kept my scores and still recommend an accept of this paper.

---

### Decision · Program_Chairs · 2025-09-17

**Decision:**

Accept (poster)

**Comment:**

The reviewers and I are in unanimous agreement that the paper makes an insightful contribution to the theory and practice of tokenization. We especially find the approach of reducing tokenization to vertex cover novel and elegant, especially in contrast to the approach of the recently-posted paper "Tokenisation is NP-Complete" [wbp24] (as pointed out by Reviewer MQ49). The method, theory, and experimental results are a clear study on the topic.

The main concerns of the papers are 1) only comparing with base BPE/Unigram rather than other variations on these, and 2) "only" evaluating on a 1B LM (and the 1B LM gains in Figure 2 and Table 4 do not seem extremely significant). The response here seems reasonable, as the partition cover approach seems novel enough and possible to augment with other variations. We encourage you to continue exploring these.